# Elastic ViTs from Pretrained Models without Retraining

Walter Simoncini[1,2] *    Michael Dorkenwald[2] *
Tijmen Blankevoort[3]    Cees G.M. Snoek[2]    Yuki M. Asano[1]

[1]University of Technology Nuremberg    [2]University of Amsterdam    [3]NVIDIA

## Abstract

Vision foundation models achieve remarkable performance but are only available in a limited set of pre-determined sizes, forcing sub-optimal deployment choices under real-world constraints. We introduce SnapViT: single-shot network approximation for pruned Vision Transformers, a new post-pretraining structured pruning method that enables elastic inference across a continuum of compute budgets. Our approach efficiently combines gradient information with cross-network structure correlations, approximated via an evolutionary algorithm, does not require labeled data, generalizes to models without a classification head, and is retraining-free. Experiments on DINO, SigLIPv2, DeIT, and AugReg models demonstrate superior performance over state-of-the-art methods across various sparsities, requiring less than five minutes on a single A100 GPU to generate elastic models that can be adjusted to any computational budget. Our key contributions include an efficient pruning strategy for pretrained Vision Transformers, a novel evolutionary approximation of Hessian off-diagonal structures, and a self-supervised importance scoring mechanism that maintains strong performance without requiring retraining or labels. Code and pruned models are available at: https://elastic.ashita.nl/

## 1 Introduction

Recent advances in model architectures and training recipes have enabled the training of vision foundation models with several billion parameters [12, 53, 16, 60], achieving state-of-the-art results across a wide range of tasks. However, these models must operate under strict compute, latency, and cost constraints when deployed in real-world settings. Yet, only a limited set of model sizes e.g., 21M, 29M, 86M, 300M, 840M and 6.7B parameter vision transformers (ViTs) for the DINOv3 family [15, 59] are made available, forcing users to select the largest model that still fits their requirements, a choice that can often be sub-optimal.

Traditionally, challenges related to model flexibility have been tackled using knowledge distillation [27]. However, this strategy mandates a predetermined target architecture and relies on often non-public pretraining datasets, which can in turn undermine robustness and limit flexibility. In parallel, methods for elastic inference [14, 7, 29, 70] have emerged to enable dynamic selection among multiple sub-networks at inference time. Yet, these methods necessitate networks designed with a predefined structure, such as nested Matryoshka [34], and require such structures to be present during pretraining. This dependency restricts their applicability to existing or proprietarily pretrained models.

An attractive alternative is structured pruning [37, 23, 24], a technique that reduces memory and computational requirements, enabling models to be adapted to diverse deployment settings. Despite their promise, most pruning techniques are tailored to specific compute constraints and tasks [35, 83]

---

* Denotes equal contribution.

39th Conference on Neural Information Processing Systems (NeurIPS 2025).

and typically require retraining, leaving a significant gap for developing universally adaptable models. To bridge this gap, we propose a novel structured pruning method, **SnapViT**, that operates in a post-pretraining setting and enables elastic inference. By that, we can extract a continuum of sub-networks from a single pretrained model, thereby enabling users to precisely tailor state-of-the-art models to their computational budget and task.

To this end, we introduce a prunability score that enables effective pruning across varying sparsity levels in a single shot that is extremely fast (less than five minutes on one A100 GPU). This score facilitates selective pruning of transformer components (e.g., row-column combinations within feed-forward blocks [14]) and larger structures, such as entire attention heads. Specifically, our prunability score is composed of two terms: (i) a gradient-based component, in line with previous works [37, 35, 80], and (ii) a novel cross-network correlation score that approximates parameter sensitivity, as captured by the off-diagonal elements of the Hessian. While prior approaches have largely ignored this component due to its quadratic scaling with the number of parameters, we propose an efficient approximation using an evolutionary algorithm to estimate these correlations. Moreover, by basing the gradient term on a self-supervised loss, our method works on any pretrained model without requiring a classification head and generalizes well across diverse downstream target datasets.

We evaluate pruned models across eight datasets, including ImageNet-1k, via k-nearest neighbor classification (k-NN), linear probing, and linear semantic segmentation using small and large ViTs from the DINO [9, 59], AugReg [61], DeIT [64, 65], and SigLIPv2 [66] model families. Our method consistently matches or outperforms state-of-the-art approaches such as LAMP [38], the LLM Surgeon [67], FPTP [35], SparseGPT [18], SNIP Magnitude [31], and NViT [76] across various sparsity levels, while generating all sparsities in a single shot. In particular, our method can prune DINOv1 ViT-B/16 to 40% sparsity, accelerating inference by 1.58x while keeping the accuracy degradation below 5%. We also show that our method is compatible with post-pruning weight correction and full fine-tuning, and outperforms or is competitive with state-of-the-art approaches in these setups. Finally, we demonstrate the importance of modeling cross-network correlations through multiple ablations and visualize the sparsity distribution across the network. Our contributions can be summarized as follows:

- We introduce an effective, fast pruning strategy for pretrained ViTs, yielding elastic models that can adapt to any computational constraints.
- We propose a novel strategy to approximate the off-diagonal components of the Hessian for network structures using a genetic algorithm.
- We obtain state-of-the-art performance under considerable pruning without retraining or requiring any labels.

## 2  Related work

**Network efficiency**   To improve model efficiency, several techniques have been proposed, including pruning [22], quantization [52], and knowledge distillation [27]. Pruning, in particular, aims at eliminating the "unimportant" bits of the network while preserving model performance. Most pruning research [25, 2, 71, 39] has focused on CNNs for image classification. Pruning methods can be classified into unstructured [22], removing individual weights to yield irregular sparsity and high compression, often requiring specialized hardware to realize speedups, or as structured [39], eliminating entire filters, channels, or other structures to enable practical acceleration on standard hardware. Finding the "unimportant" parts of the network has been done based on weight magnitudes [38, 23, 62, 44, 84, 43] activations [62, 81, 82], gradients [77, 56], or the model's Hessian [37, 24, 67, 68, 35, 63, 45, 69]. The latter is the most accurate, as it accounts for all second-order dependencies [37]; however, computing the full Hessian is infeasible. Several tractable approximations have been introduced, such as diagonal [37, 63], block diagonal [36, 18], and block diagonal with K-FAC [67, 68]. While these approximations efficiently capture local and intra-layer interactions, they inherently disregard inter-layer dependencies. To overcome this limitation, we introduce a black-box evolutionary algorithm that circumvents the need for explicitly computing the Hessian and can model intra-layer dependencies.

**Elastic inference**   The idea of extracting multiple smaller models from a single larger model has been widely explored [79, 78, 5, 20, 6], mostly in the context of CNNs. OFA [5] trains a

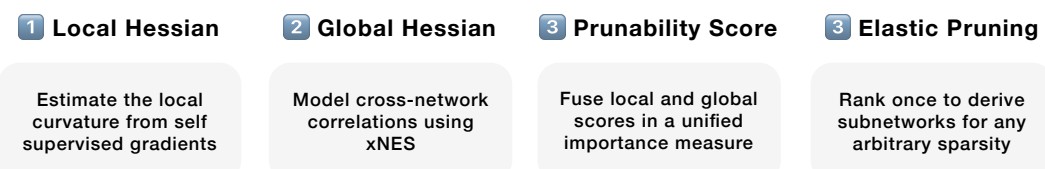

Figure 1: **Overview of our pruning method.** We decompose the Hessian into local and global components: the local Hessian is approximated from self-supervised gradients, while the global Hessian models cross-block correlations learned via an evolutionary algorithm. Combining both yields a unified prunability score that ranks parameters once, allowing single-shot generation of sub-networks at any desired sparsity. The pseudocode for our algorithm is listed in Appendix D.2.

teacher CNN model and employs distillation to fine-tune randomly sampled, non-nested submodels within a universal student model. Slimmable networks [79] jointly optimize models but offer only a limited set of predefined widths. Universal Slimmable Networks [78] extend this concept by allowing sampling from a continuous search space of submodels and jointly optimizing them. HAT [70] trains a universal network solely to learn the relative performance of different architectures; however, it requires NAS to identify the optimal architecture and trains it from scratch before serving. DynaBERT [29] jointly trains a fixed set of submodels but lacks a search strategy, limiting its approach to explicitly trained granularities. Matryoshka representations [34] can adapt to diverse downstream tasks while accommodating varying computational constraints through a nested representational structure. Various works have leveraged such a nested structure for multi-modal [7, 30], encoder-only or decoder-only [14], diffusion [21], and state-space [58] models. Notably, Matformer [14] trains transformers with this specific structure for the feed-forward part of transformer blocks from scratch, yielding a versatile model.

In contrast to these works, we explore how to derive an elastic Vision Transformer model from *any* pretrained network *without* specialized pre-training or retraining. Our method can prune both feed-forward blocks and attention heads, and requires less than five minutes on a single A100 GPU.

## 3 Method

Our method enables single-shot generation of sparse subnetworks from pretrained models, regardless of computational budget. To this end, we assign a prunability score $P$ to each network structure, e.g., row-column combinations within feed-forward blocks [14] or entire attention heads, and remove the least important ones to meet the computational constraint. Our method is summarized in Figure 1.

### 3.1 Blockwise Hessian decomposition

The importance of a parameter can be expressed by the change it induces in the objective function $\mathcal{L}$ when perturbed or removed. While directly measuring this effect is ideal [42], it is infeasible for large-scale networks with billions of parameters $N$. Following [37, 24], we approximate the loss variation under a small perturbation $\delta\boldsymbol{\theta}$ using a second-order Taylor expansion

$$\delta\mathcal{L} = \nabla_{\boldsymbol{\theta}}\mathcal{L}^{\top}\delta\boldsymbol{\theta} + \tfrac{1}{2}\delta\boldsymbol{\theta}^{\top}\mathbf{H}\delta\boldsymbol{\theta} + \mathcal{O}(\|\delta\boldsymbol{\theta}\|^3), \tag{1}$$

where $\mathbf{H}$ is the Hessian of $\mathcal{L}$ with respect to $\boldsymbol{\theta}$. Assuming the model is near a local minimum [37, 24, 35], the first-order term vanishes ($\nabla_{\boldsymbol{\theta}}\mathcal{L}^{\top}\delta\boldsymbol{\theta} \approx 0$), leaving the Hessian as the dominant indicator of sensitivity and parameter coupling. Each entry $\mathbf{H}_{ij} = \frac{\partial^2 \mathcal{L}}{\partial\theta_i\partial\theta_j}$ quantifies how parameters $\theta_i$ and $\theta_j$ interact; The off-diagonal elements thus capture correlations and redundancy across parameters.

However, computing the full Hessian is intractable since it contains $N^2$ entries. Practical approximations, such as diagonal [37], block-diagonal [35], or Kronecker-factored (KFAC) [67, 68], reduce cost but capture only local dependencies, e.g., within a single transformer block. We instead approximate the Hessian as a composition of a local term $\mathbf{H}^{(l)}$, capturing intra-block structure, and a global correlation term $\mathbf{H}^{(g)}$, which modulates sensitivities across $B$ functional units (e.g., attention heads or MLP blocks). This formulation provides a scalable, data-driven representation of both local and inter-layer dependencies without explicitly forming the full $N^2$ Hessian.

We next approximate the local curvature term $\mathbf{H}^{(l)}$ using self-supervised gradients, which yields an efficient diagonal estimate of parameter sensitivity before learning global correlations in Sec . 3.3.

## 3.2 Local Hessian approximation using SSL

We first estimate the local curvature of the loss surface following [24, 80] as

$$\boldsymbol{H}^{(l)} \approx \frac{1}{N_D} \sum_{i=1}^{N_D} \|\nabla_{\boldsymbol{\theta}} \mathcal{L}_i\|^2 \,, \tag{2}$$

computed over a dataset $D$ with $N_D$ samples while retaining only the diagonal entries of the Hessian (see Fig. 1, bottom). This diagonal approximation captures parameter-wise sensitivity within each block and provides an efficient proxy for the local curvature $\mathbf{H}^{(l)}$. To obtain gradients in a model-agnostic way, we adopt the self-supervised DINO objective [9], which removes dependence on a classification head and allows pruning of both supervised and foundation models. For an input image, we sample $n_g$ global and $n_l$ local crops, compute their normalized embeddings $z^g$ and $z^l$, and minimize the cross-view consistency loss

$$\mathcal{L}^{\text{SSL}} = \sum_{k=1}^{n_g} \sum_{m=1}^{n_l} \mathcal{L}_{\text{CE}}(z_k^g, z_m^l), \tag{3}$$

where $\mathcal{L}_{\text{CE}}$ denotes the soft cross-entropy between teacher and student embeddings.

The resulting diagonal curvature $\mathbf{H}^{(l)}$ serves as a baseline measure of local parameter sensitivity. In the next stage, xNES learns structure-wise scaling factors $\mathbf{c} \in \mathbb{R}^B$ that rescale these sensitivities across network structures based on inter-block correlations.

## 3.3 Global Hessian estimation via xNES

Even with structure-wise grouping of $\boldsymbol{H}^{(g)}$, computing all structure-level Hessian entries remains infeasible. Instead, we employ the Exponential Natural Evolution Strategy (xNES) [19] to model these interactions implicitly, without explicitly forming the Hessian. By doing so, we can *simulate pruning and measure sensitivity directly*, which has been shown to be more reliable than pure analytic approximations [42].

The diagonal Hessian estimate $\boldsymbol{H}^{(l)}$ from Sec. 3.2 provides a baseline sensitivity for each parameter. During the xNES optimization, we combine these local scores with sampled global factors $\mathbf{c} \sim \mathcal{N}(\boldsymbol{\mu}, \boldsymbol{\Sigma})$, which represent structure-wise reweightings. Each candidate $\mathbf{c}$ rescales the local sensitivities to produce a trial pruning mask; this mask is evaluated using a label-free fitness metric, allowing the covariance $\boldsymbol{\Sigma}$ to evolve toward the inverse of the true inter-block Hessian. This coupling ensures that the global correlations learned by xNES are grounded in the local curvature of the pretrained model.

We parameterize the search distribution as a multivariate Gaussian $\mathcal{N}(\boldsymbol{\mu}, \boldsymbol{\Sigma})$ over potential solutions, with an exponential parameterization of the covariance matrix $\boldsymbol{\Sigma} = \mathbf{B}\mathbf{B}^{\top}$ where $\mathbf{B} = e^{\mathbf{A}}$. xNES performs natural-gradient updates on both the mean and the covariance

$$\Delta\boldsymbol{\mu} = \eta_{\mu} \, \nabla_{\boldsymbol{\mu}}^{\text{nat}} J(\boldsymbol{\mu}, \boldsymbol{\Sigma}), \tag{4}$$

$$\Delta\mathbf{A} = \eta_{\Sigma} \, \nabla_{\mathbf{A}}^{\text{nat}} J(\boldsymbol{\mu}, \boldsymbol{\Sigma}), \tag{5}$$

where $J(\boldsymbol{\mu}, \boldsymbol{\Sigma}) = \mathbb{E}_{\mathbf{c} \sim \mathcal{N}(\boldsymbol{\mu}, \boldsymbol{\Sigma})}[F(\mathbf{v})]$ and $F$ denotes the fitness score. To measure the fitness for each sampled $\mathbf{c}$, we prune the model according to the rescaled local sensitivities and evaluate the resulting representation quality. We compute embeddings $z$ from the original model and $z_{p_s}$ from the pruned model at sparsity $s \in \mathcal{S}$, compress them via PCA to 192 dimensions, and measure cosine similarity

$$F = \frac{1}{|\mathcal{S}|} \sum_{s \in \mathcal{S}} \text{sim}(\text{PCA}(z), \text{PCA}(z_{p_s})) \,. \tag{6}$$

The natural-gradient update then adjusts $\boldsymbol{\Sigma}$ such that its inverse reflects which blocks can be pruned jointly without degrading this similarity.

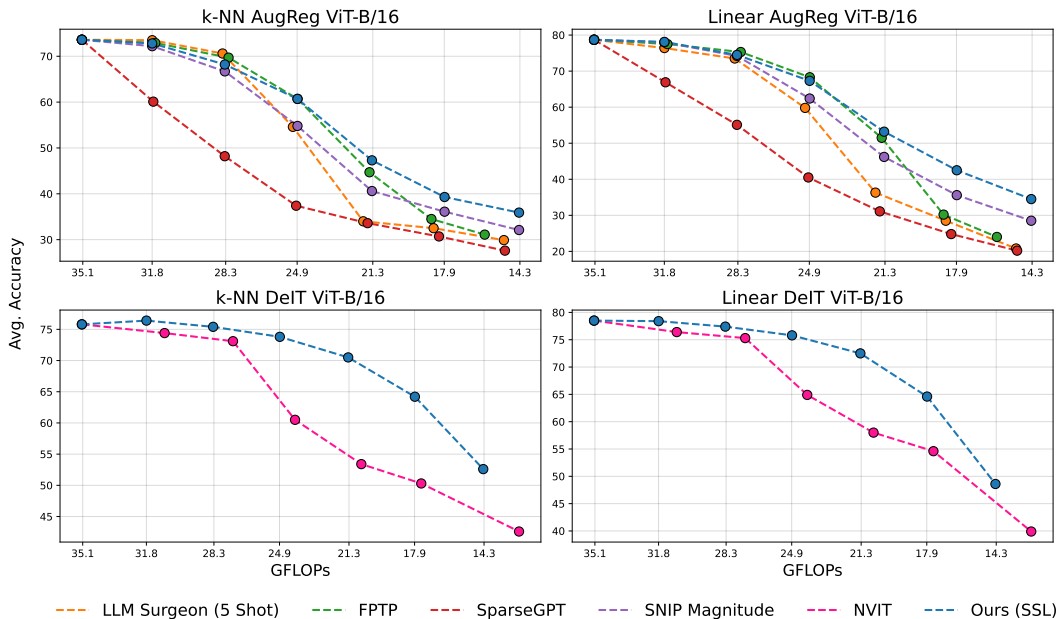

Figure 2: **Our method matches or improves upon the state-of-the-art in a retraining-free setup, while not using labels.** Top-1 accuracies in k-nearest neighbor and linear classification averaged across 7 datasets for supervised AugReg and DeIT ViT-B/16 models. Our label-free method outperforms or matches baselines that utilize labels, especially at high sparsity ratios.

We interpret the evolving covariance as a data-driven proxy for the inter-block curvature,

$$\mathbf{H}^{(g)} \approx \alpha \, \mathbf{\Sigma}^{-1}, \tag{7}$$

where $\alpha$ is a scaling constant that does not affect the relative importance of correlations. Although this is an approximation, prior analyses [57, 1] show that evolution strategies naturally adapt their covariance to the inverse Hessian on locally quadratic landscapes. In practice, the off-diagonal terms of $\mathbf{\Sigma}$ evolve to mirror cross-block dependencies, providing a tractable estimate of the global Hessian that complements the diagonal local scores.

*Intuition.* xNES contracts variance along steep directions and expands it along flat ones, so repeated updates drive $\mathbf{\Sigma}^{-1}$ to approximate the underlying curvature structure. This makes $\mathbf{\Sigma}^{-1}$ a useful Hessian surrogate for capturing both intra- and inter-block sensitivities, improving pruning robustness.

### 3.4 Elastic pretrained ViT pruning

Combining the local and global Hessian approximations yields a unified *prunability score* for each parameter. We define it as

$$\boldsymbol{P} = \mathrm{diag}\left(\frac{1}{N_D} \sum_{i=1}^{N_D} \left\| \nabla_{\boldsymbol{\theta}} \, \mathcal{L}^{\mathrm{SSL}} \right\|^2 \right) \; \odot \; \mathbf{M} \, \boldsymbol{c}, \tag{8}$$

where the diagonal term captures *local* parameter sensitivity and the blockwise scaling vector $\boldsymbol{c}$, learned through xNES, encodes *global* inter-block correlations. The membership matrix $\mathbf{M} \in \{0,1\}^{N \times B}$ expands each block factor $c_b$ to all parameters within its corresponding block, ensuring dimensional consistency with the parameter vector of size $N$. This reweighting amplifies or suppresses local sensitivities depending on the block's global importance: blocks whose parameters co-vary strongly with others (high off-diagonal curvature) receive larger effective scores, whereas isolated or redundant blocks are down-weighted.

After computing $\boldsymbol{P}$, we globally rank all parameters to determine the pruned subset at any desired sparsity level $S$

$$\Theta_S = \left\{ \theta_i \in \Theta \mid \mathrm{rank}(P_i) < |\Theta| \, (1 - S) \right\}, \tag{9}$$

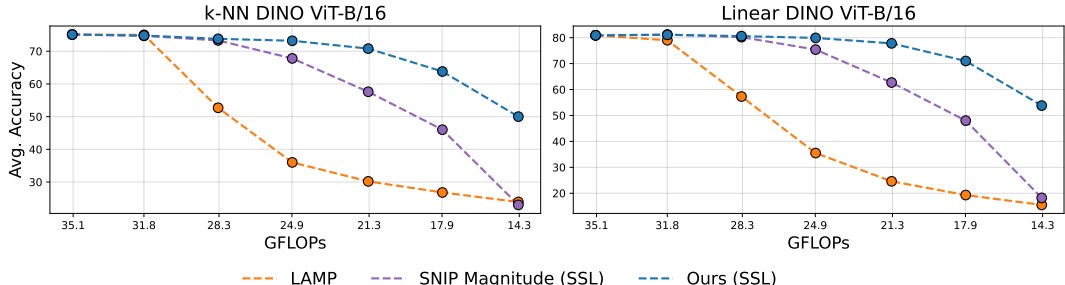

Figure 3: **Our method outperforms baselines for DINO ViT-B/16.** Top-1 accuracy in k-nearest neighbor and linear classification averaged across 7 datasets for models pruned with our method, LAMP, and SNIP Magnitude. Our method can prune DINO to 40% sparsity with an accuracy degradation under 5%.

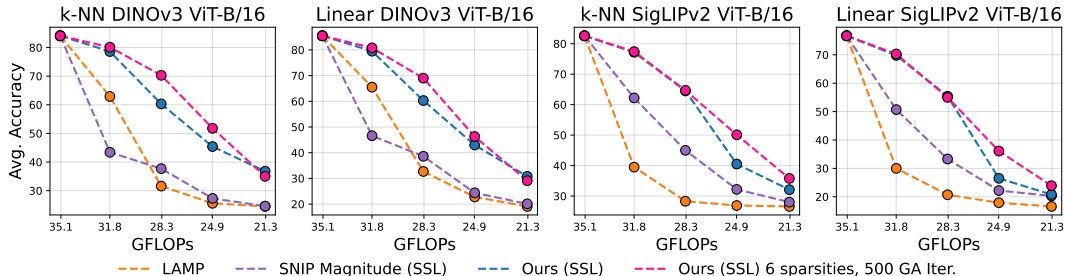

Figure 4: **Large-scale pretraining complicates pruning.** Top-1 accuracy in k-nearest neighbor and linear classification for pruned DINOv3 [59] and SigLIPv2 ViT-B/16 [66] models. We find that self-supervised models trained on large datasets are harder to prune, benefit from longer optimization horizons, and optimizing for more sparsities.

where $\Theta$ is the complete parameter set and $P_i$ denotes the score of parameter $\theta_i$. This global ranking enables **single-shot pruning**, as any target sparsity $S \in [0, 1]$ can be realized without retraining, Hessian storage, or additional optimization. The same evolutionary run therefore produces a full continuum of compute-adaptive sub-networks.

## 4 Experiments

We evaluate models pruned at six evenly spaced sparsity levels between 10% and 60%, using k-nearest neighbor (k-NN) classification, linear probing, and linear semantic segmentation. We compare our single-shot pruning protocol against state-of-the-art multi-shot approaches under the same conditions. Additionally, we assess our method with post-pruning refinements, such as weight correction and full fine-tuning, and benchmark it against prior methods in this setting. We prune hidden neurons in feed-forward blocks (equivalent to a row-column combination) and whole attention heads. We report speedups in GFLOPs. The complete experimental details are described in Appendix D. We evaluate our method on supervised DeIT [64] and AugReg [61] models and self-supervised DINO [9], DINOv3 [59] and SigLIPv2 [66] models, comparing it to state-of-the-art methods such as the LLM Surgeon [67], LAMP [38], FPTP [35], NViT [76], SparseGPT [18], and SNIP Magnitude [31].

All baselines were evaluated using their official implementation, except for LAMP and SNIP Magnitude, which we implemented in our framework. Notably, all existing methods [38, 35, 67, 76, 18, 31] optimize for a single predetermined sparsity level, whereas our approach optimizes over all sparsity levels simultaneously. We evaluate pruned models on 7 image classification datasets described in Appendix C, and report the averaged top-1 accuracy, unless otherwise specified. For k-nearest neighbor classification, we utilize the scikit-learn implementation [50] with majority voting and 20 neighbors; for linear classification, we train a linear head using stochastic gradient descent for 100 epochs, following the same recipe as DINO [8], and for linear semantic segmentation, we use the same recipe as NeCo [48].

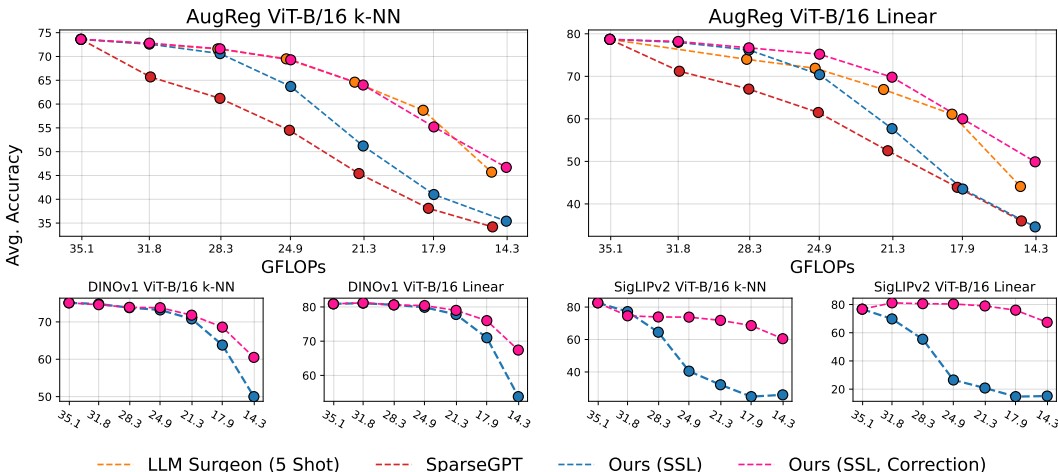

Figure 5: **Weight correction helps retaining post-pruning performance.** A single SparseGPT-style weight correction step greatly improves performance at high sparsity levels while preserving efficiency. Our method matches or surpasses state-of-the-art baselines across pruning ratios and preserves self-supervised model accuracy even under extreme sparsity (bottom row).

## 4.1 Single-Shot Structured Pruning

We evaluate our method in the *single-shot structured pruning* setting, without any post-processing such as weight correction or fine-tuning, and compare against state-of-the-art baselines. Unlike our approach, most existing methods [67, 18, 35, 75] assume the availability of a classification head; therefore, we use an AugReg ViT-B/16 backbone for a fair comparison. In addition, our method produces all sparsity levels within a single run, whereas each baseline must be executed once for every target sparsity. For NViT, we instead use a DeiT ViT-B/16 model, as their codebase is not easily extensible to other backbones. For self-supervised models, we benchmark against methods that do not require a classification head, i.e., LAMP and SNIP-Magnitude, using self-supervised gradients for the latter to ensure consistency.

**Supervised backbones** Figure 2 reports the accuracy in k-nearest neighbor and linear classification for supervised models pruned using our label-free method (Ours SSL) versus other state-of-the-art pruning techniques that make use of labels. The results show that our method can match or outperform all baselines, especially at high sparsity ratios, where it improves by 7% and 12.3% over SNIP and FPTP, respectively, at 50% sparsity. Notably, we often outperform the LLM Surgeon, which prunes models to a target sparsity in 5 shots.

**Self-supervised backbones** In Figure 3 and Figure 4, we evaluate our approach on foundation models, including DINOv1 [8], DINOv3 [59], and SigLIPv2 [66] ViT-B/16, and compare it to LAMP and SNIP-Magnitude. Since other pruning methods depend on a classification head, they cannot be applied to these models. Our method prunes DINOv1 ViT-B/16 to 40% sparsity with less than a 5% drop in accuracy, achieving a 15.1% and 53.2% improvement in linear classification over SNIP Magnitude and LAMP, respectively. We further observe that foundation models trained on large-scale datasets, such as DINOv3 and SigLIPv2, with 1.7 and 10 billion training samples, respectively, are harder to prune, benefit from longer optimization horizons (500 iterations versus a baseline of 50) and optimizing for six sparsity levels rather than four. Despite this, our method outperforms both LAMP and SNIP Magnitude, improving over the second-best method in linear classification by up to 21.7% and 34.3% for SigLIPv2 and DINOv2 ViT-B/16, respectively.

**Semantic segmentation** In Figure 6, we benchmark our method in linear semantic segmentation on Pascal VOC 2012 [17] for AugReg and DeIT ViT-B/16 backbones, reporting the best performances on the validation set. The results show that our method matches or outperforms the state-of-the-art, especially at high sparsity ratios, for example, improving by 9.1% over SNIP Magnitude at 50% sparsity for AugReg and by 15.3% over NViT at 60% sparsity for DeIT.

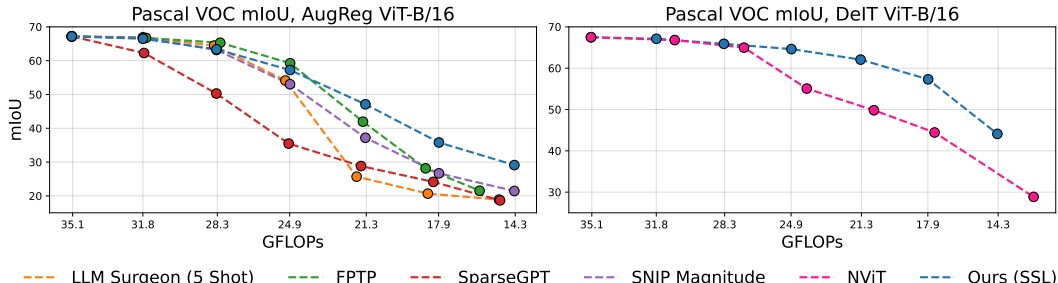

Figure 6: **Our method retains segmentation performance best.** We report the mIoU on Pascal VOC 2012 for linear semantic segmentation for AugReg and DeIT ViT-B/16 models.

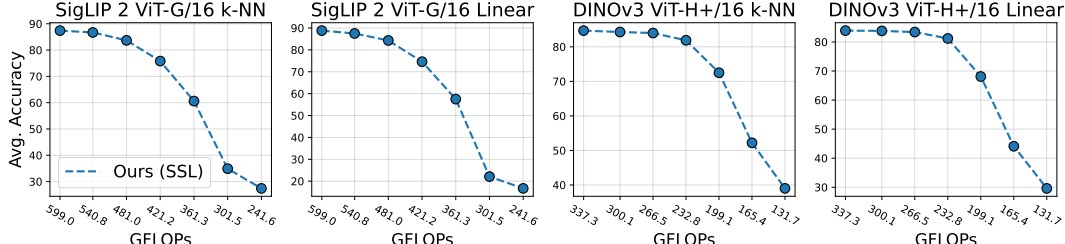

Figure 7: **Huge models are not necessarily more prunable.** Top-1 accuracy in k-nearest neighbor and linear classification averaged across 7 datasets for DINOv3 ViT-H+/16 and SigLIPv2 ViT-G/16 models pruned with our method. Contrary to a DeIT-III ViT-H/14, shown in Figure 12, performance quickly degrades beyond 30% sparsity.

## 4.2 Structured Pruning with Post-Processing

While not a core component of our method, we evaluate performance after a single SparseGPT-style weight correction step or full fine-tuning, the latter performed using the same setup as NViT. We compare the performance of our models after weight correction or fine-tuning to comparable state-of-the-art methods for supervised backbones.

**Weight Correction** We compare our approach with methods that include a weight-correction step. Note that we apply weight correction to a single sparsity and thus do not produce elastic models in this setup. We extend our method by applying a SparseGPT-style weight correction after pruning with our original algorithm. For each pruned weight matrix, we compute a *layer-wise* Hessian approximation using input activations collected from 1000 random ImageNet-1k training samples. The inverse Hessian is then used to update the remaining weights to minimize the reconstruction error of the corresponding output activations. This process is applied sequentially, layer by layer, starting from the first Transformer block. The results are shown in Figure 5, where we compare the performance of our method, with and without weight correction, to SparseGPT and the LLM Surgeon, the latter of which performs five correction steps instead of one. Our method is either competitive or outperforms the state-of-the-art at all pruning ratios. We also show that a single weight-correction step can largely preserve the performance of self-supervised models at extreme sparsity levels. In particular, we can prune SigLIPv2 to 50% sparsity with negligible performance loss in linear classification.

**Full Fine-tuning.** We also compare our method with prior pruning and model adaptation approaches that rely on extensive fine-tuning. Similar to NViT and SAViT [83], we fine-tune a DeIT ViT-B/16 pruned to 50% sparsity for 300 epochs on ImageNet-1k, using the same recipe as NViT. We compare our results to the author-reported linear classification performance on ImageNet-1k in Table 1, alongside the average linear and k-nearest-neighbor classification accuracies for open-weight models. The results show that our pruned model outperforms the unpruned model on ImageNet-1k after full fine-tuning and is competitive with the state-of-the-art, matching or outperforming it. While NViT achieves the highest ImageNet-1k accuracy, it generalizes less effectively: on average across our seven benchmark datasets, its k-nearest neighbor and linear classification performance are 1.7% and 3.9% lower, respectively, than that of a model pruned with our method and fine-tuned.

Table 1: **ImageNet-1k full fine-tuning recovers performance for 50% pruning.** Our method fully recovers pre-pruning performance on ImageNet-1k and is competitive with other state-of-the-art approaches on ImageNet-1k, while generalizing better in k-nearest neighbor and linear classification.

| Method | Avg. k-NN | Avg. Linear | ImageNet-1k | Fine-tuning Epochs |
|---|---|---|---|---|
| Unpruned | 75.8 | 78.5 | 81.8 | – |
| SN-Net [46] | 70.2 | 71.4 | 80.0 | 100 |
| NViT [75] | 73.7 | 72.0 | 83.3 | 300 |
| LPViT [74] | – | – | 80.6 | 300 |
| SAViT [83] | – | – | 82.6 | 300 |
| SnapViT (Ours) | 75.4 | 75.9 | 82.6 | 300 |

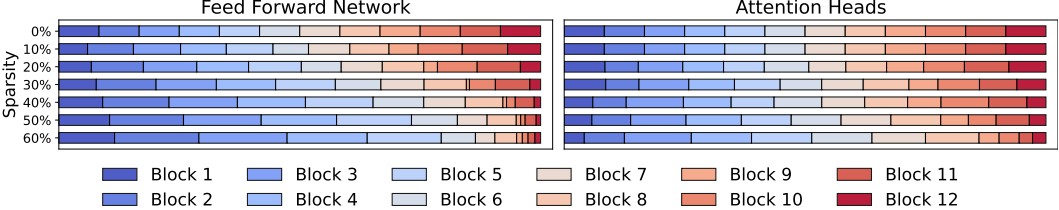

Figure 8: **Deeper blocks are heavily pruned in DINO ViT-B/16:** visualization of the normalized parameter allocation for DINO ViT-B/16 models pruned at increasing sparsity levels (0-60%). Feed-forward blocks, especially deep ones (8-12), are pruned before attention heads, while earlier blocks maintain their parameter density, revealing the network's inherent structural redundancy.

## 4.3 Pruning big ViTs

Given the efficiency of our method, which only approximates the Hessian, we can apply it to prune large ViTs. As shown in Figure 7, we prune SigLIPv2 ViT-G/16 and DINOv3 ViT-H+/16 models, containing 1.2B and 840M parameters, respectively. While both models retain performance up to 30% sparsity, their accuracy drops sharply beyond this point. In contrast, our results on DeIT-III ViT-H/14 (Figure 12) show stable performance for up to 50% sparsity. We hypothesize that this difference arises from the pretraining regime: DeIT-III is pretrained on ImageNet-21k (13M samples), whereas SigLIPv2 and DINOv3 are pretrainedit on substantially larger datasets with 10B and 1.7B samples, respectively. Large-scale pretraining likely distributes representational knowledge more evenly across parameters, making it less obvious which units can be pruned. Nonetheless, combining our method with simple weight correction techniques can recover performance even for models that undergo large-scale pretraining, as demonstrated in Figure 5 for SigLIPv2 ViT-B/16.

## 4.4 Ablations

**Sparsity allocation**   In Figure 8, we visualize the sparsity allocation across blocks for DINO ViT-B/16. The visualization reveals two key patterns: (i) pruning initially favors slimming feed-forward blocks, leaving attention heads largely intact, though they too undergo pruning at higher sparsity ratios, but to a lesser extent; (ii) blocks 8 to 12 demonstrate higher pruning susceptibility, suggesting these layers contain more redundant information. As sparsity increases, pruning progressively concentrates in these blocks while earlier blocks maintain a relatively stable parameter density.

**Importance of global interactions.**   Table 3 illustrates the impact of approximating the global Hessian $H^{(g)}$ using more cross-network interactions. In particular, we compare the average k-NN accuracy of DINO ViT-B/16 models pruned to 50% sparsity while modeling either no interactions ($B = 0$), equivalent to not using the genetic algorithm, only the interactions between feed-forward blocks ($B = 12$), and interactions between all pairs of feed-forward blocks and attention heads ($B = 156$). The results show that the performance of pruned models increases as more global interactions are modeled, by up to +6.9% in average k-nearest neighbor accuracy.

**Sparsity-specific vs continuous optimization.**   In Figure 9, we compare the performance of models pruned with our method while optimizing for each individual target sparsity against our one-shot

Table 2: **Performance improves with more genetic algorithm iterations.** Average accuracy in k-NN and linear classification versus the number of iterations for an AugReg ViT-B/16 backbone pruned to 50% sparsity.

| Iterations | Avg. k-NN | Avg. Linear |
|---|---|---|
| 50 | 39.3 | 42.2 |
| 250 | 39.9 | 42.2 |
| 500 | **40.9** | **44.0** |

Table 3: **Modeling more cross-network interactions improves performance.** Average top-1 accuracy in k-NN versus the interactions modeled and optimized using the genetic algorithm, for a DINO ViT-B/16 backbone pruned to 50% sparsity.

| Interactions Modeled | Avg. k-NN |
|---|---|
| None (0) | 56.6 |
| FFN (12) | 60.1 |
| FFN and heads (156) | **63.5** |

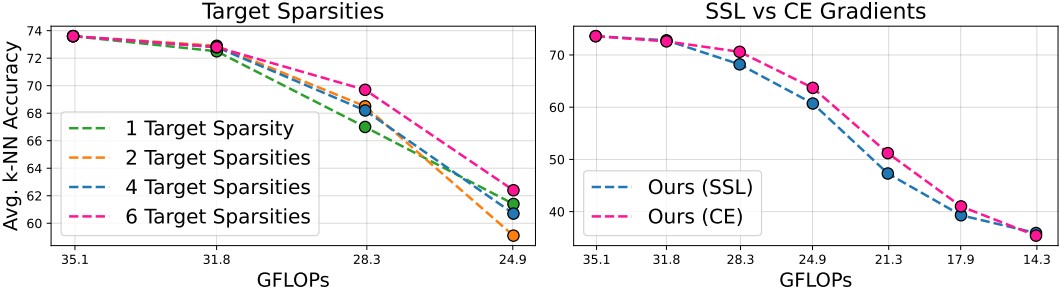

Figure 9: **Ablations for the number of target sparsities and loss.** Average accuracy in k-nearest neighbor classification for AugReg ViT-B/16 models pruned by optimizing for 1 to 6 sparsities (left), and for models pruned using gradients from either a self-supervised or cross-entropy loss (right).

to elastic model approach, and show that both produce models with similar performance, with the advantage that our approach only needs to be run once. Furthermore, we compare optimizing the genetic algorithm for two, four, and six sparsities, and find that optimizing for more sparsities can significantly improve performance.

**Function evaluations.** In Table 2 we ablate the number of iterations used for the genetic algorithm optimization and show that running the algorithm for 500 iterations improves the performance of an AugReg ViT-B/16 pruned to 50% sparsity by up to 1.8% on average in linear classification.

**Supervised gradients.** In Figure 9 we ablate the choice of a self-supervised loss to guide pruning, and observe that, for an AugReg ViT-B/16 backbone, using a cross-entropy loss only performs marginally better.

## 5   Conclusion

In this work, we presented a novel and fast post-training structured pruning method that enables elastic inference across a continuum of sparsity levels. Our approach combines gradients and cross-structure correlations, approximated via a genetic algorithm, to produce efficient ViTs with strong performance across several tasks without retraining. Furthermore, we have shown that it is possible to effectively prune models without requiring labeled data or a classification head via a self-supervised loss.

**Acknowledgment**   This work has received financial support from Qualcomm Technologies Inc., the University of Amsterdam, and the Top Consortia for Knowledge and Innovation (TKIs) allowance from the Netherlands Ministry of Economic Affairs and Climate Policy. The authors gratefully acknowledge the scientific support and HPC resources provided by the Erlangen National High Performance Computing Center (NHR@FAU) of the Friedrich-Alexander-Universität Erlangen-Nürnberg (FAU) under the BayernKI project v115be. BayernKI funding is provided by Bavarian state authorities.

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

## A  Broader impact

Our method accelerates the inference speed of vision transformers, reducing the computing requirements and power usage of these models. Thus, our method could have positive consequences, such as lowering the $CO_2$ emissions generated by inference and enabling users with limited compute resources to benefit from the abilities of large (pruned) models. We believe our method should not have a direct negative impact.

## B  Limitations

We have shown that our approach can yield elastic models in a label- and retraining-free fashion that perform on par or better than the current state of the art, especially at high sparsity ratios, but some limitations remain:

- Self-supervised models trained on large-scale datasets are difficult to prune effectively, and performance quickly collapses. Yet, we have found that longer optimization horizons and initializing $\Sigma$ with cross-structure CKA scores, as shown in Appendix G, can improve performance. Thus, we believe that there is a way to prune these models effectively, but we leave this for future work.

- The re-weighting of prunability scores alters the ranking of attention heads and MLP blocks, but does not take into account individual MLP hidden neurons. Optimizing the ranking of individual neurons could further enhance the performance of sparse models.

## C  Data and models

We investigate the performance of pruned models on 7 image classification datasets, namely ImageNet-1k [55], FGVC Aircraft [41], Oxford-IIT Pets [49], DTD Textures [11], EuroSAT [26] and CIFAR 10/100 [33], plus Pascal VOC 2012 [17] for semantic segmentation. Table 4 lists all the datasets used in this paper alongside their license and citation.

We follow the standard evaluation protocol for each individual dataset and report the top-1 accuracy in k-nearest neighbor and linear classification for image classification datasets and the mean intersection over union (mIoU) for Pascal VOC.

We use the train/test splits defined by the dataset authors where possible, except for EuroSAT, for which we use an 80/20 stratified split as indicated by the dataset paper. We always report the performance on the test split, except for ImageNet-1k and Pascal VOC, for which we report performance on the validation split. For the linear classification experiments we use the validation split defined by the dataset authors if available, and otherwise create one using an 80/20 random split.

Table 4: **Datasets.** Summary table of the datasets used in the paper.

| Dataset | License | Citation |
|---|---|---|
| ImageNet-1k | Research Only | [55] |
| FGVC Aircraft | Research Only | [41] |
| Oxford-IIT Pets | CC BY-SA 4.0 | [49] |
| DTD Textures | Research Only | [11] |
| EuroSAT | MIT | [26] |
| CIFAR 10/100 | Unknown | [33] |
| Shaders21k | Unknown | [4] |
| DiffusionDB | CC0 1.0 | [72] |
| Pascal VOC 2012 | Unknown | [17] |

Table 5 lists the models pruned and evaluated in this paper, alongside their citation and license.

Table 5: **Models.** Summary table of the models used in the paper.

| Model | License | Citation |
|---|---|---|
| DINO | Apache 2.0 | [9] |
| DINOv3 | DINOv3 License | [59] |
| SigLIPv2 | Apache 2.0 | [66, 73] |
| AugReg | Apache 2.0 | [61, 73] |
| DeIT | Apache 2.0 | [64] |
| DeIT-III | Apache 2.0 | [65] |

Table 6: **Our algorithm scales sub-linearly with respect to the number of parameters.** Runtime, as measured on an NVIDIA A100, of our algorithm for the entire DeiT-III family.

| Model | Layers | Attention Heads | Parameters (M) | Runtime |
|---|---|---|---|---|
| DeIT-III ViT-S/16 | 12 | 6 | 22.1 | 2m 35s |
| DeIT-III ViT-B/16 | 12 | 12 | 86.6 | 2m 55s |
| DeIT-III ViT-L/16 | 24 | 16 | 304.4 | 4m 58s |
| DeIT-III ViT-H/14 | 32 | 16 | 632.1 | 11m 4s |

# D  Experimental details

## D.1  Complexity Analysis

The total computational cost of our pruning algorithm is

$$\mathcal{O}(N_D F) \; + \; \mathcal{O}(T\lambda S N_S F') \; + \; \mathcal{O}(TB^2) \; + \; \mathcal{O}(P \log P), \tag{10}$$

where

- $\mathcal{O}(N_D F)$ computes the local diagonal Hessian via one forward-backward pass on $N_D$ samples, where $F$ denotes the cost per pass.

- $\mathcal{O}(T\lambda S N_S F')$ covers the xNES search phase: in each of $T$ iterations, $\lambda$ candidates are drawn and each is evaluated across $S$ sparsity targets using $N_S$ images. $F'$ represents a forward-only pass (feature extraction + PCA), requiring no back-propagation.

- $\mathcal{O}(TB^2)$ accounts for updating the $B \times B$ covariance matrix $\Sigma$ in xNES, which models global off-diagonal dependencies between functional blocks (e.g., FFN and attention heads).

- $\mathcal{O}(P \log P)$ sorts the $P$ prunability scores once to obtain elastic subnetworks for any desired sparsity level.

Compared with SOSP [45] and EigenDamage [69], our approach eliminates all explicit curvature computations. SOSP-H requires one Hessian-vector product per structure, and SOSP-I constructs a dense $S \times S$ Gauss-Newton matrix. In contrast, our algorithm performs only forward inference and a lightweight covariance update $\mathcal{O}(TB^2)$, avoiding any backward curvature passes.

The runtime is thus dominated by the forward feature-extraction term. In practice, with $T = 50$ xNES iterations and four sparsity targets, a single A100 GPU prunes the entire DeiT-III family in only a few minutes, as shown in Table 6. The quadratic covariance term remains negligible even for the largest models, demonstrating excellent scalability and making our approach one of the most efficient second-order-aware pruning frameworks to date.

## D.2  Pseudo Code

Algorithm 1 outlines our single-shot pruning procedure. Given a model $f_\theta$, a dataset $D$, a maximum number of iterations $T$, a set of target sparsities $\mathcal{S}$ and a population size $\lambda$, which we initialize as $\lambda = 4 + 3\log(d)$ as described in the xNES paper [19], where $d$ is the problem dimensionality, e.g., 156 in the case of a ViT-B/16, we proceed as follows:

1. Compute the self-supervised gradients, obtain the local prunability scores, and initialize the xNES mean $\mu$ and covariance $\Sigma$.

**Algorithm 1** Single-shot pruning with xNES

---

**Require:** Model $f_\theta$; dataset $D$; blocks $\{1, \ldots, B\}$; iterations $T$; population $\lambda$; sparsity grid $\mathcal{S}$
**Ensure:** Global ranking / masks for arbitrary sparsity

1: $\mathbf{s} \leftarrow \mathrm{diag}\Big(\frac{1}{N_D} \sum_{x \in D} \|\nabla_\theta \mathcal{L}^{\mathrm{SSL}}(x)\|^2\Big)$      ▷ dataset-averaged diagonal Hessian proxy

2: $(\boldsymbol{\mu}, \boldsymbol{\Sigma}) \leftarrow (\mathbf{0}, \mathbf{I})$      ▷ xNES mean & covariance

3: **for** $t = 1$ to $T$ **do**

4:     **for** $k = 1$ to $\lambda$ **do**

5:        $\mathbf{c}^{(k)} \sim \mathcal{N}(\boldsymbol{\mu}, \boldsymbol{\Sigma})$      ▷ blockwise reweighting factors

6:        $\mathbf{u}^{(k)} \leftarrow (\mathbf{M}\,\mathbf{c}^{(k)}) \odot \mathbf{s}$      ▷ expand to parameters and combine with local scores

7:        **for** $s \in \mathcal{S}$ **do**

8:           $\mathrm{mask}^{(k,s)} \leftarrow \mathrm{TopK}\big(\mathbf{u}^{(k)}, \mathrm{budget}(s)\big)$

9:           $z \leftarrow f_\theta(x), \quad z_{p_s} \leftarrow f_{\theta \odot \mathrm{mask}^{(k,s)}}(x)$      ▷ forward-only on a minibatch $x$

10:          $F^{(k,s)} \leftarrow \cos\big(\mathrm{PCA}(z), \mathrm{PCA}(z_{p_s})\big)$

11:        **end for**

12:        $F^{(k)} \leftarrow \frac{1}{|\mathcal{S}|} \sum_{s \in \mathcal{S}} F^{(k,s)}$

13:     **end for**

14:     $(\boldsymbol{\mu}, \boldsymbol{\Sigma}) \leftarrow \mathrm{XNES\text{-}UPDATE}\big(\{\mathbf{c}^{(k)}, F^{(k)}\}_{k=1}^\lambda\big)$

15: **end for**

16: $\mathbf{c}_{\mathrm{final}} \leftarrow \arg\max_k F^{(k)}$      ▷ best-performing sample (global correlation vector)

17: $\boldsymbol{P} \leftarrow (\mathbf{M}\,\mathbf{c}_{\mathrm{final}}) \odot \mathbf{s}$      ▷ final prunability scores

18: **return** $\mathrm{argsort}(\boldsymbol{P})$      ▷ derive masks for any target sparsity by thresholding

---

2. Sample $\lambda$ individuals for the current generation. For each individual, combine the local and global prunability scores, produce the pruning masks for each target sparsity $s \in \mathcal{S}$, and, for each sparsity $s$, measure the fitness as the average post-PCA cosine similarity between pruned and original embeddings. The individual's fitness $F$ is then computed as the average of fitnesses across the sparsity targets $\mathcal{S}$.

3. Update $\mu$ and $\Sigma$, and continue from step 2.

The algorithm terminates after $T$ steps, and the best ranking is derived from the individual with the highest fitness.

### D.3 Pruning

We prune models to six target sparsities, namely 10, 20, 30, 40, 50, and 60% in one shot. To do so, we first estimate gradients using either a DINO or a cross-entropy loss and 1000 random samples from the ImageNet-1k training set (unless specified otherwise) and batch size 16. Gradients are averaged over each batch and summed across batches. We do not use any data augmentation for the cross-entropy loss, and for the DINO loss, we only use random cropping to generate 2 global and 10 local crops, with scales between $(0.25, 1.0)$ and $(0.05, 0.25)$, respectively.

After approximating the gradients, we compute prunability scores using Equation 2, and produce a single score for each attention head and hidden feed-forward neuron by averaging. Then, we optimize the sparsity allocation using the xNES for 50 iterations. For each iteration, we generate models pruned at 10, 30, 50, and 60% sparsity and measure the cosine similarity between embeddings produced by the pruned models and the original model, using 1000 fixed samples from the ImageNet-1k training set. We then average the cosine similarity across sparsity ratios and select the configuration that maximizes this metric and, by consequence, minimizes divergence. Before computing the cosine similarity, we project embeddings to 192 dimensions using a PCA model trained using the same 1000 images, as embedded using the original model.

For each block, we constrain our algorithm to prune at most 80% of the attention heads and 95% of the feed-forward neurons, leaving at least 2 attention heads and 154 neurons for each individual block in the case of a ViT-B/16 model.

## D.4 Post-pruning processing

**Weight correction.** We apply SparseGPT-style post-pruning weight correction to models pruned with our method as follows: first, we apply our pruning algorithm to obtain a binary mask $M$ for a given target sparsity $s$, where $M_{i,j} = 0$ indicates that the weight at position $(i, j)$ is pruned. Then, for each layer to be pruned, we collect $N$ input activations in a matrix $X \in \mathbb{R}^{d_{\text{in}} \times N}$ and compute the damped Hessian as

$$H = XX^T + \lambda I, \qquad \lambda = \frac{0.01}{d_{\text{in}}} \sum_{i=1}^{d_{\text{in}}} H_{i,i}. \tag{11}$$

We then invert the Hessian using a Cholesky decomposition, obtaining $H^{-1}$. Following SparseGPT, we then process the weight matrix column-wise in blocks of $B = 128$ columns. For each column $j$ in a block $i : i + B$, we mask out the pruned weights and compute the reconstruction error as:

$$E_{:,j-i} = (1 - M_{:,j}) \odot \frac{W_{:,j}}{[H^{-1}]_{j,j}}. \tag{12}$$

We then update the unpruned weights of subsequent columns as

$$W_{:,j:(i+B)} = W_{:,j:(i+B)} - E_{:,j-i} \cdot H^{-1}_{j,j:(i+B)}. \tag{13}$$

After applying the weight correction to all pruned layers in a Transformer block, we rearrange the attention heads and MLP hidden neurons according to our ranking and remove the pruned structures as in our retraining-free experiments.

**Full fine-tuning.** We fine-tune a DeIT ViT-B/16 model pruned to 50% sparsity with our method and a self-supervised loss using the fine-tuning scripts from NViT, closely following their recipe. In particular, we fine-tune the model in float16 for 300 epochs, using 8 GPUs, a per-device batch size of 144, and an initial learning rate of 0.0002. We use hard distillation with $\alpha = 0.5$ and soft distillation with $\tau = 20.0$ from a RegNetY-16GF [51] teacher. We combine the two losses as $\mathcal{L} = \mathcal{L}_{\text{hard}} + 10000\mathcal{L}_{\text{soft}}$.

## D.5 Baselines

**LLM Surgeon [67].** We adapt the official implementation[1], released under the BSD 3-Clause Clear License, to prune ViTs, disable weight correction and LoRA fine-tuning and closely follow the configuration recommended by the paper authors, except for the number of samples used to estimate the curvature (1000 versus the default 128 to match our method) and the number of shots, 5 in our experiments compared to the recommended 40, as we did not notice significant differences in our preliminary runs. While we prune the hidden neurons of feed-forward blocks and whole heads, the LLM Surgeon prunes independent rows and columns in weight matrices, making a 1:1 comparison hard, as it has more degrees of freedom compared to our method. Moreover, due to their pruning strategy, the speed improvements of the LLM Surgeon are not easy to realize in practice.

**NViT [76].** We evaluate NViT using the code and configuration available in the official GitHub repository[2], released under the NVIDIA Source Code License-NC, with the exception that we prune 1024 structures per step rather than 32 due to computational reasons. In contrast to other methods, we do not disable fine-tuning during pruning, as doing so causes the algorithm to fail. For the full fine-tuning experiment, we evaluate the fine-tuned checkpoint made available by the authors.

**FPTP\* [35].** We adapt the official code implementation[3] to ViTs, disable mask tuning and prune models using the default parameters, except for the number of samples used for estimating gradients, for which we use 1000 instead of the standard 2048 for a fair comparison with other methods.

---

[1] https://github.com/Qualcomm-AI-research/llm-surgeon
[2] https://github.com/NVlabs/NViT
[3] https://github.com/WoosukKwon/retraining-free-pruning

Table 7: **Theoretical FLOPs formulas for ViTs.** ViT components, sub-components, individual computations, and the formula used to estimate the corresponding theoretical FLOPs. The formula to estimate the FFN FLOPs assumes that the hidden dimensionality is $4d_{\text{model}}$.

| Component | Sub-Component | Computation | FLOPs |
|---|---|---|---|
| Embeddings | – | – | $2n_{\text{patch}}d_{\text{patch}}^2 n_{\text{channels}}d_{\text{model}}$ |
| Logits | – | – | $2d_{\text{model}}n_{\text{classes}}$ |
| Block | Attention | QKV | $2n_{\text{tokens}}3d_{\text{model}}(d_{\text{key}}n_{\text{heads}})$ |
| | | QK Logits | $2n_{\text{tokens}}^2(d_{\text{key}}n_{\text{heads}})$ |
| | | Softmax | $3n_{\text{heads}}n_{\text{tokens}}^2$ |
| | | Reduction | $2n_{\text{tokens}}^2(d_{\text{key}}n_{\text{heads}})$ |
| | | Projection | $2n_{\text{tokens}}(d_{\text{key}}n_{\text{heads}})d_{\text{model}}$ |
| | FFN | – | $16n_{\text{tokens}}d_{\text{model}}^2$ |

**SNIP Magnitude**[*] **[31].** We implemented the SNIP Magnitude score in our framework following its official implementation[4]. Score aggregation and pruning are done in the same way as for our method.

**SparseGPT [18].** We adapt the official code implementation[5], released under an Apache 2.0 License, to ViTs and to perform structured pruning by masking entire columns. We use the default parameters, except for the number of samples used to estimate the Hessian, which is set to 1000 for a fair comparison with other methods.

**LAMP [38].** We implemented the LAMP score in our framework, closely following the formulas and pseudocode from the original paper. We aggregate scores and perform pruning as for our method.

## D.6  Evaluation

**k-nearest neighbor classification.** We evaluate pruned models in k-nearest neighbor classification using the implementation from scikit-learn [50]. In particular, we report the classifier performance using majority voting across 20 neighbors and $L_2$-normalized features.

**Linear classification.** We evaluate pruned models in linear classification following the DINO recipe [9]. For each dataset, we train a linear classification head for 100 epochs using SGD with a 0.9 momentum, a learning rate of 0.001, no weight decay, a batch size of 256, and a cosine annealing learning rate scheduler [40] with $\eta_{\text{min}} = 0$. We then select the best classifier on the validation set and report its performance on the test set. No data augmentation is applied to the training samples.

**Semantic segmentation.** We evaluate models in semantic segmentation on Pascal VOC 2012 [17] by training a convolutional head for 25 epochs using SGD with a 0.9 momentum, a 0.01 learning rate, further reduced to 0.001 after 20 epochs, a 0.0001 weight decay, and a batch size of 128 following the recipe from [48]. We select the best model on the validation set, and reports its average mean intersection over union (mIoU). Training images are augmented via random crops with a scale between 80 and 100% of the original image, resized to $(224, 224)$, and flipped horizontally with a 50% chance.

## D.7  GFLOP definition

We use the term **GFLOPs** to indicate the number of theoretical floating-point operations required for a single forward pass. We adopt the formulas from [28, 10], displayed in Table 7, where $n_{\text{patch}}$ indicates the total number of patch tokens for an input image (e.g., 196 assuming a patch size of 16 and an input size of $224 \times 224$), $d_{\text{patch}}$ the side length of a single image patch, $n_{\text{channels}}$ the number of channels of the input image (e.g., 3 for a RGB image), $d_{\text{model}}$ the embedding size, $n_{\text{tokens}}$ the total

---

[4] https://github.com/tuna0724/Pruning
[5] https://github.com/IST-DASLab/sparsegpt
[*] The official code implementation was released without an explicit license.

Table 8: **GFLOPs for ViT models.** Theoretical GFLOPs measurements for ViT models of increasing size and their architectural configuration parameters that contribute to the computation.

| Model | $n_{\text{patch}}$ | $d_{\text{patch}}$ | $n_{\text{ch}}$ | $d_{\text{model}}$ | $n_{\text{tokens}}$ | $d_{\text{key}}$ | $n_{\text{layers}}$ | $n_{\text{heads}}$ | $n_{\text{classes}}$ | GFLOPs |
|---|---|---|---|---|---|---|---|---|---|---|
| ViT-S/16 | 196 | 16 | 3 | 384 | 197 | 64 | 12 | 6 | 1000 | 9.2 |
| ViT-B/16 | 196 | 16 | 3 | 768 | 197 | 64 | 12 | 12 | 1000 | 35.1 |
| ViT-L/16 | 196 | 16 | 3 | 1024 | 197 | 64 | 24 | 16 | 1000 | 123.2 |

Table 9: **The image classification evaluation has high variance at high sparsity ratios.** Average top-1 k-nearest neighbor and linear classification accuracies and their standard deviation across three seeds (0, 13, and 42) using an AugReg ViT-B/16 backbone pruned to 10, 20, and 30% sparsity.

| Eval. | Sparsity | DTD | FGVC | EuroSAT | CIFAR 10 | CIFAR 100 | Pets | IN1K | Avg. |
|---|---|---|---|---|---|---|---|---|---|
| k-NN | 10% | $61.0 \pm 0.7$ | $24.7 \pm 0.6$ | $91.7 \pm 0.4$ | $92.7 \pm 0.3$ | $73.5 \pm 0.2$ | $89.7 \pm 0.2$ | $76.6 \pm 0.4$ | $72.8 \pm 0.1$ |
| | 30% | $51.2 \pm 1.8$ | $18.3 \pm 1.1$ | $91.4 \pm 0.6$ | $76.8 \pm 3.5$ | $49.3 \pm 3.3$ | $70.9 \pm 6.5$ | $55.4 \pm 2.8$ | $59.0 \pm 2.0$ |
| | 50% | $37.1 \pm 2.0$ | $11.5 \pm 1.8$ | $89.7 \pm 1.5$ | $56.8 \pm 0.4$ | $29.9 \pm 0.3$ | $28.3 \pm 5.2$ | $22.9 \pm 2.3$ | $39.5 \pm 1.4$ |
| Linear | 10% | $72.0 \pm 0.3$ | $38.4 \pm 0.7$ | $94.2 \pm 0.2$ | $93.6 \pm 0.3$ | $80.2 \pm 0.1$ | $92.2 \pm 0.0$ | $77.3 \pm 0.2$ | $78.3 \pm 0.1$ |
| | 30% | $62.9 \pm 1.4$ | $29.4 \pm 1.1$ | $92.2 \pm 0.1$ | $79.8 \pm 3.6$ | $58.3 \pm 3.0$ | $81.2 \pm 4.1$ | $60.6 \pm 1.7$ | $66.3 \pm 1.5$ |
| | 50% | $52.2 \pm 2.3$ | $18.1 \pm 2.1$ | $84.7 \pm 1.9$ | $51.5 \pm 0.8$ | $21.6 \pm 2.4$ | $43.7 \pm 5.3$ | $20.9 \pm 1.9$ | $41.8 \pm 1.2$ |

number of tokens including the `[CLS]` token (and distillation token for DeITs), $d_{\text{key}}$ the attention head size, $n_{\text{classes}}$ the number of output units for the classification head, $n_{\text{layers}}$ the number of transformer blocks and $n_{\text{heads}}$ the number of attention heads. The total number of FLOPs is computed as:

$$\text{Total FLOPs} = \text{embeddings} + n_{\text{layers}} \cdot (\text{attention} + \text{FFN}) + \text{logits}. \tag{14}$$

For a ViT-B/16, this results in approximately 35.1 GFLOPs. Some papers report multiply-accumulate operations (MACs) instead of FLOPs, equivalent to FLOPs/2, i.e., 17.6 GMACs for a ViT-B/16. Theoretical measurements for other model sizes, alongside the relevant architectural details, are illustrated in Table 8.

## E   Statistical significance of results

In Table 9, we report the mean accuracy and one standard deviation computed across three seeds (0, 13, 42) in k-nearest neighbor and linear classification for AugReg ViT-B/16 models pruned using our method to 10, 30 and 50% sparsity. Accuracies for models pruned to 10% sparsity are consistent across seeds. In contrast, sparser models have higher standard deviations on average and on certain datasets, such as Oxford-IIT Pets, on which the k-nearest neighbor accuracy at 30% sparsity has a standard deviation of 6.5%.

## F   Compute resources

The pruning experiments were run using a NVIDIA A100 GPU with 40GB of VRAM, 16 CPU cores, and 40 GB of RAM. While the pruning runtime is negligible, evaluating each (model, sparsity) pair in k-nearest neighbor and linear classification requires approximately one GPU hour in float16. Given this, we estimate that reproducing the main experiments presented in this paper would require approximately 275 GPU hours, covering the one-shot pruning experiments (with and without weight correction) and all ablations. The full fine-tuning experiment required an additional 2.5 days on 8 GPUs (480 GPU hours), bringing the total compute budget to roughly 755 GPU hours. Preliminary exploratory runs required less than 50 GPU hours in total.

## G   Additional experimental results

**Importance of data.**   We ablate the pruning performance with respect to the data used to estimate gradients and for the genetic algorithm optimization for DINO ViT-B/16. In particular, we compare DiffusionDB [72], a dataset of synthetic images generated via Stable Diffusion [54], Shaders 21K [4], a dataset of abstract images generated via shared programs, ImageNet-1k [13] which was used for pretraining and strongly aligns with some of the evaluation datasets, including ImageNet-1k itself,

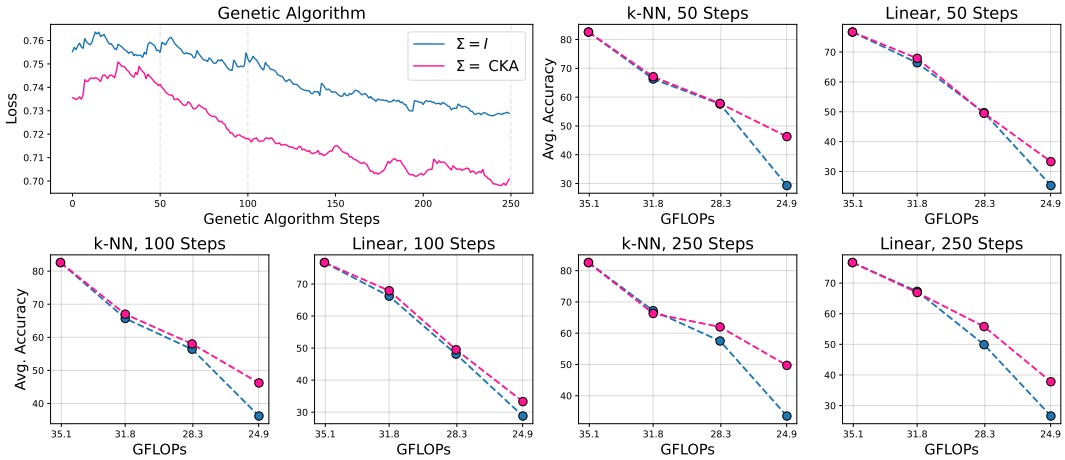

Figure 10: **Initializing $\Sigma$ using CKA scores improves performance for SigLIPv2 ViT-B/16.** The genetic algorithm loss over 250 steps, smoothed using an exponential moving average with $w = 0.95$, when $\Sigma$ is initialized either with an identity matrix or the CKA scores between structures, plus the model performance when pruned to up to 30% sparsity after 50, 100 and 250 genetic algorithm steps.

Table 10: **The CKA initialization significantly improves performance across datasets.** Top-1 accuracy in k-nearest neighbor and linear classification of SigLIPv2 ViT-B/16 pruned to 30% sparsity using our method with either an identity or CKA initialization for $\Sigma$ and 250 genetic algorithm steps.

|  | Initialization | DTD | FGVC | EuroSAT | CIFAR 10 | CIFAR 100 | Pets | IN1K |
|---|---|---|---|---|---|---|---|---|
| k-NN | $\Sigma = I$ | 51.4 | 13.3 | 85.1 | 49.0 | 24.5 | 43.0 | 35.8 |
| | $\Sigma = $ CKA | **63.5** | **24.6** | **88.2** | **75.7** | **48.0** | **60.1** | **56.9** |
| Linear | $\Sigma = I$ | 4.9 | 1.0 | 74.8 | 40.4 | 18.5 | 38.2 | 33.9 |
| | $\Sigma = $ CKA | **11.3** | **2.3** | **76.7** | **68.3** | **41.9** | **58.3** | **54.9** |

CIFAR 10 and 100 [33] and Oxford-IIT Pets [49], and a dataset obtained by sampling a total of 1000 images in equal parts from the training split of each of the evaluation datasets, which we call "Merged Data". The results, shown in Figure 11, demonstrate that alignment between pruning and task data heavily affects performance, improving by up to 6.4% when comparing ImageNet-1k-based pruning to Shaders-21k at 40% sparsity. Furthermore, while the Merged Data dataset performs similarly to ImageNet-1k at shallow sparsity ratios, it can improve by up to 5.3% at 60% sparsity, suggesting that a data-centric view of pruning can help produce sparse models that generalize better.

**Genetic algorithm initialization.** By default, xNES initializes $\Sigma = I$, where $I$ is the identity matrix. We compare this strategy to initializing each entry $\Sigma_{i,j}$ using the centered kernel alignment (CKA) [32] score between the activations of structures $i$ and $j$ (e.g. two attention heads, one attention head and a feed-forward block or two feed-forward blocks), estimated using 2500 random samples from the training set of ImageNet-1k for a SigLIPv2 ViT-B/16 backbone. We run the genetic algorithm for up to 250 steps for both initializations and evaluate the performance of pruned models at 50, 100, and 250 steps. The results, shown in Figure 10, demonstrate that the CKA initialization improves both convergence and performance, as the initial loss is lower, and a significant gap persists even after 250 steps. Regarding performance, the CKA initialization matches or outperforms the baseline across all pruning ratios at 50, 100, and 250 steps, with a gap of up to 16.4% in k-nearest neighbor at 250 steps and 30% sparsity. Table 10 reports the model performance at 30% sparsity on a per-dataset basis, showing that a CKA initialization can improve performance by up to 26.7% in k-nearest neighbor and 27.9% in linear classification on CIFAR 10.

**Pruning across model sizes.** In Figure 12, we plot the average accuracy in k-NN and linear classification across the seven image classification datasets for DeIT-III [65] models, trained using ImageNet-22k and fine-tuned on ImageNet-1k, ranging from ViT-S/16 to H/14, pruned to up to

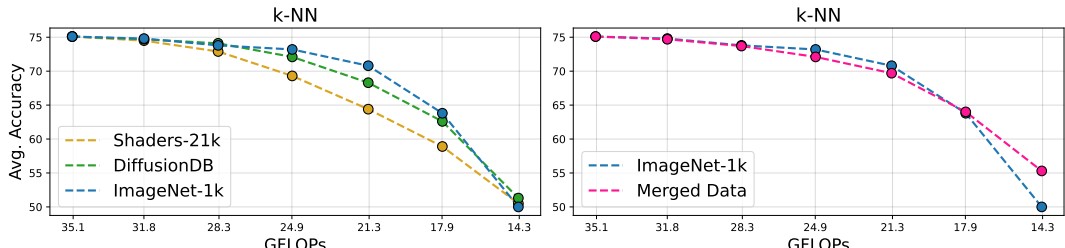

Figure 11: **Alignment between pruning data and target tasks improves performance.** Average top-1 accuracy in k-nearest neighbor and linear classification of DINO ViT-B/16 models pruned using our method and data from ImageNet-1k, DiffusioDB, Shaders 21k, and a 1000-samples dataset built by sampling equally from the training set of each of the evaluation datasets, named "Merged Data".

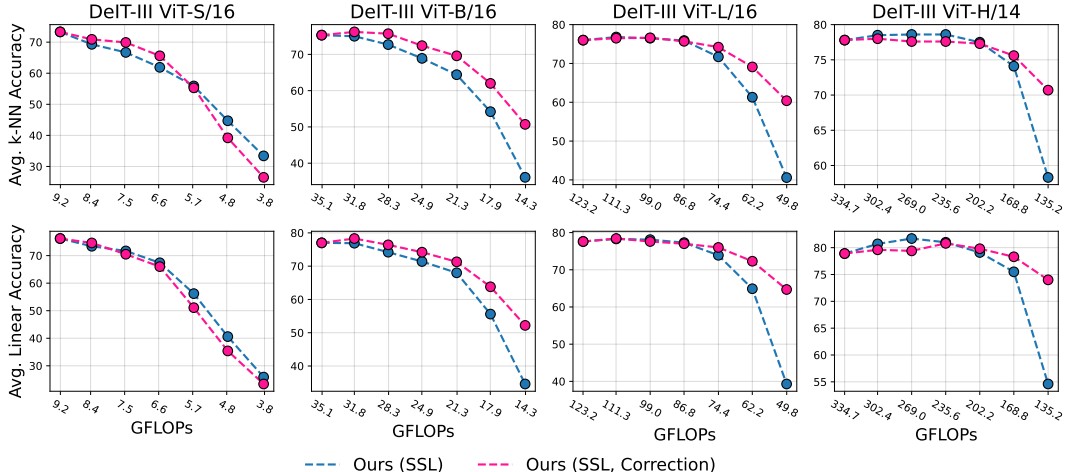

Figure 12: **Larger supervised models can be pruned more aggressively.** Top-1 accuracy in k-nearest neighbor and linear classification, averaged across 7 datasets, for models of various sizes belonging to the DeIT-III [65] family, pruned to up to 60% sparsity using our method with and without weight correction. Weight correction consistently improves performance for ViT-B/16 and larger models, by up to 25.4% for a ViT-L/16 pruned to 60% sparsity.

60% sparsity using our method. We find that larger models from this family can be pruned more aggressively with a minimal loss in performance. For example, the ViT-H/14 model can be pruned to 50% sparsity, equivalent to removing approximately 316M parameters, while losing only 3.7% and 3.4% on average in k-nearest neighbor and linear classification, respectively. The results on a per-dataset basis are shown in Table 11, where we observe that for some datasets, such as EuroSAT and Oxford-IIT Pets, the performance drop remains below 1.5% for both linear and k-nearest neighbor classification. Interestingly, accuracy even improves on FGVC Aircraft. Finally, the model maintains strong performance on ImageNet-1k, with at most a 7.6% decrease in accuracy. When post-pruning weight-correction is applied, performance is mostly restored, with an average degradation of only 0.5% in k-nearest neighbor, and an average improvement of 0.9% in linear classification at 50% sparsity. On a per-dataset basis, DTD Textures, FGVC Aircraft, and ImageNet-1k benefit the most from weight correction, improving by 7%, 5.8%, and 2.4%, respectively, in linear classification. Interestingly, weight correction improves performance for all models except for the ViT-S/16 at high pruning ratios. We hypothesize this might be due to the limited remaining representational capacity of the model, as a ViT-S/16 pruned to 50% sparsity has only 11M remaining parameters.

**Dense representation quality.** In Figure 13, we qualitatively analyze the quality of dense representations for pruned DINO ViT-B/16 models via the HummingBird in-context semantic segmentation evaluation [3], as implemented in [47], for Pascal VOC 2012. We use the default parameters except for the memory bank size and input image size, which are $196 \times 10^4$ and $224 \times 224$, respectively,

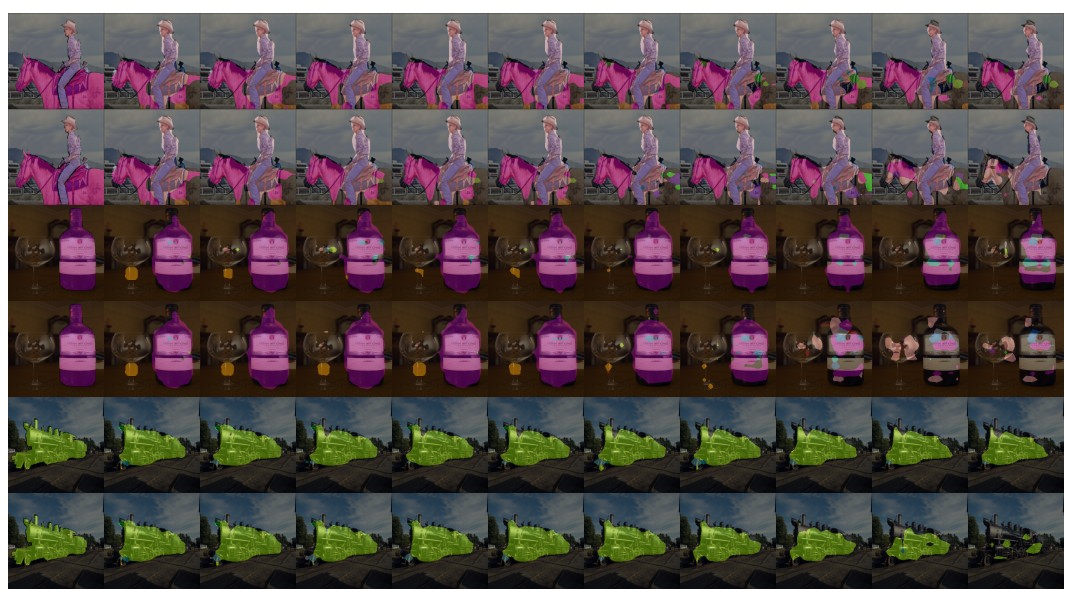

Figure 13: **Our method preserves the quality of dense representations.** Visualization of segmentation masks for Pascal VOC 2012 produced via in-context semantic segmentation using DINO ViT-B/16 models pruned from 0 to 60% sparsity using our method (top rows) and SNIP Magnitude (bottom rows). The ground truth is shown in the left-most image.

in our experiments. We prune models using our method and SNIP Magnitude, and visualize results for 10 linearly spaced pruning ratios between 0 and 60% sparsity. Similarly to the results for global understanding tasks shown in Figure 3, the representations of models pruned with SNIP Magnitude start to collapse at 40% sparsity, while representations of models pruned with our method are more robust, producing sensible segmentation masks even at 60% sparsity.

Table 11: **DeIT-III ViT-H/14 retains performance across datasets at 50% sparsity.** Top-1 accuracy in k-nearest neighbor and linear classification of a DeIT-III ViT-H/14 model pruned to 50% sparsity with our method using a SSL loss, with and without weight correction, versus the original model.

| Eval. | Sparsity | Corr. | DTD | FGVC | ESAT | CIFAR 10 | CIFAR 100 | Pets | IN1K |
|-------|----------|-------|------|------|------|----------|-----------|------|------|
| k-NN  | 0%       | –     | 62.4 | 30.1 | 89.9 | 96.8     | 86.1      | 92.4 | 86.6 |
|       | 50%      | ✗     | 58.9 | 35.5 | 88.6 | 91.9     | 73.2      | 91.7 | 79.0 |
|       | 50%      | ✓     | 60.7 | 35.0 | 90.8 | 93.3     | 74.8      | 92.3 | 82.0 |
| Linear| 0%       | –     | 69.2 | 22.0 | 92.9 | 97.4     | 90.2      | 93.9 | 86.5 |
|       | 50%      | ✗     | 60.9 | 26.5 | 93.0 | 94.4     | 79.6      | 93.4 | 80.4 |
|       | 50%      | ✓     | 67.9 | 32.3 | 93.7 | 95.2     | 82.3      | 93.8 | 82.8 |

