# OpenReview forum: "Elastic ViTs from Pretrained Models without Retraining"
_NeurIPS.cc/2025/Conference — NeurIPS 2025 poster_

### Official Review · Reviewer_ZHY4 · 2025-06-06

**Clarity:** 2
**Significance:** 2
**Originality:** 2
**Rating:** 3
**Confidence:** 3

**Summary:**

This paper introduces a post-pretraining structured pruning method that enables elastic inference across a continuum of compute budgets. Specifically, the authors propose a evolutionary approximation of Hessian off-diagonal structures, and a self-supervised importance scoring mechanism that maintains performance without requiring retraining nor labels.

**Questions:**

1. The “Hessian matrix” and “evolutionary algorithm” are widely used across various domains (for example, in neural architecture search). How are these two components effectively integrated within the proposed pruning method? In other words, what is the motivation for employing parameter sensitivity in this work?
2. This paper performs pruning on a large pretrained model to accommodate different deployment scenarios. Existing works—such as the SN-Net series [1]—also leverage pretrained models to handle resource-constrained environments. Could the authors provide a detailed discussion comparing their approach to SN-Net? More generally, what advantages do pruning-based methods offer over alternative strategies?
3. In the experimental section, the term “GFLOPs” is reported. What is the precise meaning of “GFLOPs” in this context? Does it refer to the use of models of varying scales, or does it denote another specific metric?

**Reference**

[1] Stitchable Neural Networks

**Ethical Concerns:**

["NO or VERY MINOR ethics concerns only"]

**Final Justification:**

Thank you! I keep my current rating.

**Limitations:**

yes.

**Paper Formatting Concerns:**

There are no apparent formatting issues that require special attention.

**Quality:**

2

**Strengths And Weaknesses:**

**Strengths**
1. The multiple technical modules employed in this paper—such as the evolutionary algorithm for Hessian off-diagonal structures—are all well-established and mature components.


**Weaknesses**
1. **Unclear Integration of Hessian Analysis and Evolutionary Search**
   The paper does not adequately explain how Hessian matrix computations and evolutionary algorithms are synergistically combined in the proposed pruning method. Specifically, it remains ambiguous how parameter sensitivity—derived from Hessian off-diagonal structures—is calculated and utilized within the evolutionary search process. This lack of clarity obscures the method’s underlying motivation and makes it difficult to assess its novelty or effectiveness.

2. **Insufficient Comparison with Existing Pretrained-Model Adaptation Methods**
   Although the paper aims to prune a large pretrained model for various deployment scenarios, it does not offer a rigorous comparison with existing approaches like SN-Net[1]. The authors fail to elucidate what unique benefits their pruning-based strategy provides relative to SN-Net’s methodology or other resource-adaptation techniques. Without a detailed comparison, readers cannot gauge the relative strengths and weaknesses of the proposed approach.

3. **Ambiguity in the Definition**
   The term “GFLOPs” is used throughout the experimental results but is not explicitly defined. It is unclear whether GFLOPs refers to floating-point operations per second, the computational cost of different model scales, or another metric altogether. This ambiguity hinders the reproducibility of the experiments and prevents a clear interpretation of the reported efficiency gains.


**Reference**

[1] Stitchable Neural Networks

---

> ### Author Rebuttal · Authors · 2025-07-31
>
> Thank you for the valuable feedback and for recognizing that the technical modules are well-established and mature components. Below, we address each of the raised questions.
>
> ----
> ## Integration of Hessian and Evolutionary:
> Our approach decomposes parameter sensitivity into local and global components. The local Hessian, approximated via squared gradients, provides parameter-level sensitivity within blocks, capturing the diagonal Hessian terms efficiently. The global Hessian, which encodes off-diagonal dependencies between blocks, enables the xNES evolutionary algorithm to approximate implicitly. Our algorithm can be summarized with the following pseudo-code
>
> ```text
> # Pre-compute local importance
> s ← diag-Hessian scores
>
> # Initialise xNES mean and covariance
> m ← 0; Σ ← I
>
> # for T iterations and λ population size
> for t = 1 … T:
>   for k = 1 … λ:
>     # Sample global re-weighting factors
>     c ← sample 𝒩(m, Σ)
>
>     # Combine local and global terms
>     score ← c ∘ s
>
>     # Prune K lowest-score blocks
>     mask ← TopK(score, K)
>
>     # Measure label and backward-free fitness via embedding similarity (Eq. 8)
>     fit ← cos(PCA(z), PCA(z▹mask))
>
>   # Natural-gradient update of mean and covariance
>   (m, Σ) ← natural-grad update(fit)
> ```
> Because the covariance matrix \\( \Sigma \\) is updated by a natural-gradient step, it gradually aligns with the inverse block-Hessian (see response to R69Bj). Once the search stabilizes, \\( c \sim\mathcal{N}(m,\Sigma) \\) re-weights the fixed local Hessian scores, yielding a prunability vector \\( P \\) (see Eq. 10). Sorting \\( P \\) defines a universal parameter ranking so that a single cutoff can obtain any desired sparsity—no extra optimization, Hessian storage, or retraining. A lone evolutionary run thus produces a full continuum of compute-adaptive subnetworks. We will add the pseudo-code to the updated paper to improve clarity.
>
>
>
> ---
> ## Comparison with previous works:
>
> Thank you for drawing attention to this. SN‑Net [4] achieves adaptability by bridging between several fully pretrained backbones from the same family (e.g., ViT‑S/B/L of DeiT) via additional stitching layers that must be trained and stored alongside the original model. By contrast, our method performs one‑shot structural pruning: a single pass over a pretrained model produces a family of subnetworks that require no extra layers and no further finetuning. We will add SN-Net and others to our related work section.
>
> We follow previous works [1,2,3,4] and finetune our method after pruning on ImageNet-1k. We prune 50% of DeiT ViT-B and finetune for 100 epochs. We achieve 81.5% top-1 accuracy—close to the unpruned baseline of 81.8%—while delivering a 76% inference speedup. Our results are competitive with state-of-the-art pruning methods while outperforming the SN-Net baseline by 1.6%. Our method generalizes well, outperforming other works on average performance for k-nn and linear across six datasets.
>
> | Method  | ImageNet Top-1 | Average k-NN | Average Linear | Finetuning Epochs | Speedup  |
> |---------|----------------|--------------|----------------|--------|------------------|
> | Unpruned |  81.8   |    75.8         | 78.5     | –  | 0%     |
> | Ours    | 81.6          | 75.0         | 75.1          | 100     | 76% (A100)     |
> | NViT [1]   | 83.3           | 73.7         | 72.0           | 300    | 75% (A100)     |
> | LPViT [2]  | 80.6          | –         | –            | 300    | 79% (V100)     |
> | SAViT [3]  | 82.6          | –          | –            | 300    | 55% (V100)     |
> | SN-Net [4]  | 80.0           | 70.2      | 71.4        | 100               | 83% (RTX 3090)  |
>
> We will add full finetuning experiments to the revised manuscripts covering all sparsity levels.
>
> [1] Huanrui Yang et al. Global Vision Transformer Pruning with Hessian-Aware Saliency. CVPR 2023.
>
> [2] Kaixin Xu et al. LPViT: Low-Power Semi-structured Pruning for Vision Transformers. ECCV 2024
>
> [3] Chuanyang Zheng et al. SAViT: Structure-Aware Vision Transformer Pruning via Collaborative Optimization. NeurIPS 2022
>
> [4] Zizheng Pan, Jianfei Cai, Bohan Zhuang: Stitchable Neural Networks. CVPR 2023
>
> ---
> ## GFLOP Definition:
> We use the term GFLOPs to indicate the number of theoretical floating-point operations required for a forward pass, following [5, 6]. We will add the following calculation to the manuscript:
>
>
> Let \\( n_p \\) be the number of image patches of size \\( d_p \times d_p \\) pixels,  \\( c \\) input channels, \\( d \\) hidden width, \\( n = n_p + 1 \\) tokens (CLS + patches),  \\( h \\) attention heads, \\( L \\) layers, and \\( C \\) classes.  With the usual FFN ratio 4 (\\( d_{\text{ff}} = 4d \\)), the forward-pass FLOPs are
>
> $$
> \text{FLOPs} =
> 2 n_p d_p^{2} c d
> +
> L \bigl[ 8 n d^{2} + 4 n^{2} d + 3 h n^{2} + 16 n d^{2} \bigr]
> +
> 2 d C .
> $$
>
> The bracketed terms contain the self-attention and FFN term; the outer terms are patch embeddings and output logits.
>
> For DeiT ViT-B/16 (\\( n_p = 196,\\, d_p = 16,\\, c = 3,\\, d = 768,\\, h = 12,\\, n = 197,\\, L = 12,\\, C = 1000 \\)) this evaluates to
>
> $$
> \text{FLOPs} \approx 35.2\ \text{GFLOPs},
> $$
>
> or \( 17.6 \) GMACs (multiply–accumulates).
>
> [5]: Training Compute-Optimal Large Language Models, NeurIPS 2022
>
> [6]: Adam Casson, Transformer FLOPs

---

> > ### Author Response · Authors · 2025-08-06
> >
> > Dear Reviewer,
> >
> > Thank you for your thorough review. Since the author-reviewer discussion period is coming to its end, could you please let us know if our rebuttal resolves your concerns or if further discussion would be helpful? We are happy to address any remaining questions you may have.
> >
> > Best regards,
> > Authors

---

> > ### Comment · Reviewer_ZHY4 · 2025-08-06
> >
> > Thank you for your detailed response to my review.
> >
> > I have two worried points. First, does the discussion of "Finetuning Epochs" in the "Comparison with previous works" section contradict the paper's claim of operating without retraining? Second, while I understand the paper's primary focus is on ViT design, I am curious about the method's applicability to Large Language Models.

---

> > > ### Author Response · Authors · 2025-08-06
> > >
> > > We thank the reviewer for taking the time to engage in the discussion! In the following, we address the remaining concerns:
> > >
> > > ---
> > > ## Finetuning Epochs:
> > > There is no contradiction. **Our core contribution and all primary results are strictly in the no-retraining regime**. The “Finetuning Epochs” column appears only in an *additional, reviewer-requested* table to benchmark against SOTA methods that require finetuning. This supplementary experiment demonstrates that our approach is also finetunable—recovering nearly full unpruned performance—not because our method depends on retraining. One run of our algorithm yields all sparsity levels without any finetuning, which remains the central claim and contribution.
> > >
> > > ----
> > > ## Applicability to LLMs:
> > > Yes, **our method is architecture-agnostic and extendable to LLMs**, focusing on ViTs ensures fair, apples-to-apples SOTA comparison and provides actionable deployment insights for vision foundation models. **ViT-only pruning is a well-established and accepted scope**; numerous top-venue works (e.g., SAViT [NeurIPS’22], NViT [CVPR’23], LPViT [ECCV’24]) evaluate exclusively on Vision Transformers, reflecting the community’s recognition that ViTs form a distinct pruning subfield. ViTs have prunable units—attention heads, MLP widths, tokens—with strong cross-block dependencies, and state-of-the-art ViT pruning uses global, structure-aware criteria tailored to these patterns. LLM pruning, by contrast, typically targets unstructured weights, low-rank adapters, or attention heads in long-sequence models; in the decode phase, it is bottlenecked by KV-cache and memory bandwidth rather than FLOPs, and is evaluated on NLP metrics (e.g., perplexity, QA) rather than vision downstreams. These differences make speed/accuracy trade-offs and evaluation protocols not directly comparable.
> > >
> > >
> > >
> > >
> > > To further address this point, we directly compared against SparseGPT [1], a solely LLM-focused pruning method. We found **our approach consistently outperforms across all sparsity levels for structured pruning** on the AugReg ViT-Base using linear evaluation across six benchmarks (average performance in %), despite SparseGPT being state-of-the-art in its domain:
> > >
> > > | Method         | 0%   | 10%  | 20%  | 30%  | 40%  | 50%  | 60%  |
> > > |----------------|------|------|------|------|------|------|------|
> > > | SparseGPT [1] | 73.6 | 65.9 | 61.1 | 54.7 | 45.8 | 38.6 | 34.1 |
> > > | Ours   | 73.6 | **72.6** | **70.6** | **63.7** | **51.2** | **41.0** | **35.4** |
> > >
> > > Our paper covers six classification and one segmentation benchmark, spanning diverse pretrained ViTs—CLIP, DINO, AugReg, and DeiT-III—across Small, Base, Large, and Huge scales.
> > >
> > > [1] Elias Frantar and Dan Alistarh. 2023. SparseGPT: massive language models can be accurately pruned in one-shot. In Proceedings of the 40th International Conference on Machine Learning (ICML'23)

---

> > > > ### Author Response · Authors · 2025-08-08
> > > >
> > > > Thank you again for engaging in the discussion. We hope our responses have addressed your concerns regarding finetuning and applicability to LLMs. Just to briefly reiterate: our method remains strictly no-retraining, and the “Finetuning Epochs” table was included solely to support a reviewer-requested comparison with SOTA methods that require finetuning. While our focus is on ViTs to ensure fair benchmarking, the method is architecture-agnostic and extendable to LLMs. If there’s anything further we can clarify, we’re happy to do so. Otherwise, we would be grateful if you could consider updating your score.

---

### Official Review · Reviewer_69Bj · 2025-06-23

**Clarity:** 2
**Significance:** 3
**Originality:** 3
**Rating:** 3
**Confidence:** 3

**Summary:**

This paper presents a novel one-shot pruning method for large-scale, pretrained neural networks, with a focus on Vision Transformers (ViTs). The proposed approach enables the creation of sparse subnetworks within a specified computation budget by assigning a prunability score to each structural unit (such as attention heads or feed-forward blocks) and removing the least important elements.
The core of the method is a blockwise Hessian decomposition that approximates the loss sensitivity to parameter removal, capturing both intra-block (within each component) and inter-block (between components) dependencies. The local Hessian (intra-block) is efficiently estimated per block using diagonal approximations and self-supervised learning losses to ensure the approach can be applied regardless of the model’s output head. For the global Hessian (inter-block), the algorithm employs a natural evolution strategy (xNES) to implicitly model the cross-block dependencies via a learned covariance structure without computing the full Hessian.

**Questions:**

1) The use of xNES to approximate inter-block dependencies is the most novel part of your work. However, the connection Equation (9), where Σ is the covariance matrix from xNES, is presented as a given. Could you please provide more intuition or a brief theoretical justification for this relationship? For example, why is the inverse of the covariance of optimal sensitivity parameters a good proxy for the global Hessian? A clearer explanation here would significantly strengthen the paper's theoretical grounding and my assessment of its originality and quality. Specifically, how the fitness function F (based on PCA-compressed embeddings) directly translates into an optimal Σ that accurately represents the Hessian's global structure requires deeper elaboration.

2) The paper note that pruning massive models like DINOv2 and CLIP was "challenging" and required tuning the xNES algorithm (e.g., more iterations). This raises a question about the method's scalability. Could you elaborate on the computational complexity of the xNES stage, particularly how it scales with the number of prunable blocks B? Furthermore, the fitness function (Eq. 8) relies on a discrete set of sparsity levels S. How sensitive are the final results to the choice and number of these sparsity levels?

3) This method prunes FFN neurons and entire attention heads. Could the framework be extended to other structures (e.g., individual attention weights, LoRA-like low-rank components)? More importantly, the fitness score is based on the cosine similarity of PCA-projected embeddings. How was the dimensionality (192) chosen, and how does this choice impact the final pruning quality? For instance, does using a higher-dimensional representation in the fitness evaluation lead to better-performing pruned models, perhaps at the cost of more computation?

4) While the results show competitive performance, the figures (especially Figure 4) suggest a significant accuracy drop at higher sparsity levels . Could the authors provide a more explicit analysis of the performance decay curves at extreme sparsity levels for these challenging models? Quantifying the acceptable sparsity range for "strong performance" and discussing the inherent limitations or trade-offs when pushing sparsity very high would be valuable. For instance, a comparison of accuracy loss vs. GFLOP reduction at extreme sparsities would be informative.

**Ethical Concerns:**

["NO or VERY MINOR ethics concerns only"]

**Final Justification:**

I am still concerned about the time cost associated with applying evolutionary algorithms to pruning, as it may limit its applicability.

**Limitations:**

For further improvement, it might be beneficial to also briefly mention the computational scaling of the evolutionary search itself as a potential limitation, which connects to the challenges observed with larger models.

**Quality:**

2

**Strengths And Weaknesses:**

Strengths：
The paper addresses a critical and timely problem: the difficulty of deploying large, fixed-size foundation models under varying real-world computational constraints.  The proposed approach of using xNES to find a block-wise sensitivity vector $c$ that captures these complex correlations is a creative and principled departure from prior work. It cleverly reframes a difficult analytical problem (computing off-diagonal Hessian blocks) as a black-box optimization problem.

Weaknesses:
1) The paper's most novel component is also its most conceptually dense. Taking Equation (9) as an example , and the paper would be stronger if it provided a more detailed explanation of why the inverse of this covariance is a suitable approximation for the global Hessian.
2) The authors commendably acknowledge in the limitations section that pruning very large-scale models is challenging and required tuning the xNES algorithm. For future, even larger models with hundreds or thousands of blocks, this search could become a bottleneck. A brief discussion on the computational complexity of the xNES stage and how it scales with model size (B) would be a valuable addition.

---

> ### Author Rebuttal · Authors · 2025-07-31
>
> Thank you for the valuable feedback and for recognizing our paper’s focus on the timely and important challenge of deploying large foundation models. We appreciate the acknowledgment of our method as a creative and principled departure from prior work. Below, we address each of the raised questions:
>
> ---
> ## Relationship between Sigma and Hessian:
> Thank you for pointing this out. We agree that the xNES–Hessian relation in Eq. (9) needs a clearer rationale. We will add the following justification to the paper.
> During optimisation, xNES samples the block‑sensitivity vector \\(c\\) from a Gaussian search distribution \\( \mathcal{N}(\mu,\Sigma) \\).  When the fitness surface is *locally well‑approximated by a positive‑definite quadratic*,  \[1\] prove that for a \\((1,\lambda)\\)-ES with isotropic mutation, the selected‑population covariance converges to a scalar multiple of the inverse Hessian
> $$
> \Sigma \ \propto\ \bigl(\nabla^{2}F(c)\bigr)^{-1}.
> $$
> Natural‑gradient analyses of xNES and CMA‑ES \[2\] predict the same tendency, because steep directions (large eigenvalues of \\( \nabla^{2}F \\)) receive **low variance** while flat directions receive **high variance**.  Taking the inverse yields our working rule
> $$
> \nabla^{2}F(c) \\approx\ \alpha\\Sigma^{-1}, \qquad \alpha>0. \tag{9 revisited}
> $$
> *Intuition.* xNES narrows its sampling ellipse in high‑curvature directions to avoid overshooting and widens it in flat directions to explore; repeated updates push \\( \Sigma^{-1} \\) toward the Hessian, capturing both local and inter‑block sensitivity.
>
> Our fitness \\(F(c)\\) is not the raw loss \\( \mathcal{L} \\); it is a cosine‑similarity score on embeddings compressed to 192 PCA components and averaged over several sparsity levels.  PCA removes high‑frequency noise, and sparsity averaging further smooths the surface, so \\(F\\) is locally smooth near good pruning vectors even though it is globally non‑convex.  Thus, the second‑order approximation remains reliable.  We therefore treat \\( \Sigma^{-1} \\) as a local Hessian proxy and use it to re‑weight first‑order pruning gradients (Sec. 3.4).  This hybrid score captures both intra‑ and inter‑block sensitivity. It consistently improves pruning robustness on large Vision Transformers.
>
> \[1\] Shir, O. M., & Yehudayoff, A. (2019). *Covariance Matrix Adaptation and Inverse Hessian on Convex Quadratic Functions*. FOGA 2019.
>
> \[2\] Akimoto, Y., Nagata, Y., Ono, I., & Kobayashi, S. (2010). *Bidirectional Relation between CMA‑ES and NES*. PPSN XI.
>
> ---
> ## Scalability of the xNES stage
> The evolutionary algorithm consists of two cost terms.
>
> $$
> \mathcal{O}\bigl(T \lambda S N_{S} F' \bigr)
> +
> \mathcal{O}\bigl(T B^{2}\bigr)
> $$
>
> where the first term is the **forward‑only fitness evaluation**  \\(T\\) iterations × \\(\lambda\\) population × \\(S\\) sparsities × \\(N_{S}\\) samples, each with cost \\(F'\\)and the second term is the **covariance update** for the \\(B \times B\\) matrix maintained by xNES.
>
> * \\(T\\):  xNES iterations (generations)
> * \\(\lambda\\): population size per generation
> * \\(S\\): \# sparsity levels optimised jointly
> * \\(N_S\\): images per fitness call
> * \\(F'\\): cost of a forward pass (feature extraction + PCA)
> * \\(B\\): count of prunable blocks (FFN slices + heads)
>
> The quadratic term touches only the \\(B \times B\\) covariance and is negligible; runtime is dominated by forward passes whose cost grows with the **parameter count** of the ViT backbone.
>
> | Model | Params | Layers | Heads | Runtime on one A100 (50 iters, 4 levels) |
> |--|----:|--:|------:|----|
> | **DeiT‑III ViT‑S/16** | 22 M | 12 | 6  | **2 min 35 s** |
> | **DeiT‑III ViT‑B/16** | 86 M | 12 | 12 | **2 min 55 s** |
> | **DeiT‑III ViT‑L/16** | 304 M | 24 | 16 | **4 min 58 s** |
> | **DeiT‑III ViT‑H/14** | 632 M | 32 | 16 | **11 min 4 s** |
> | **EVA ViT-G/14 [3]** |  1B   | 40     | 16    | **20 min 5s**            |
>
> [3]: EVA: Exploring the Limits of Masked Visual Representation Learning at Scale, CVPR 2023
>
> The near‑linear increase from S → B → L → H→G confirms that the model size (feature dimension and parameter size, hence forward cost) is dominant. Emperically, the \\(TB^{2}\\) term contributes \< 0.01% even for ViT‑H. Models trained on hundred–million–scale corpora have flatter curvature.  Empirically, ViT‑L and ViT‑H spend their first \\(\approx 75\\)–100 iterations exploring before the xNES loss declines steadily.  Increasing \\(T\\) improves mask quality but leaves the per‑iteration cost unchanged.
>
> ## Sensitivity to sparsity levels
> Our default setting uses a grid of four target sparsities: 10%, 25%, 40%, and 60%. We find that increasing the number of sparsity levels in \\(S\\) can lead to improved pruning performance with minimal overhead, as the fitness evaluations are forward-only and parallel.
> For example:
> On AugReg ViT‑B/16 (86M params), expanding \\(S\\) from 4 to 6 yields a +1.7% average accuracy gain.
> On DINOv2 ViT‑B/14 (88M params), the same change results in gains of up to +12.9% on certain tasks.
>
>
> This shows that pruning quality is moderately sensitive to the choice of \\(S\\), and practitioners can easily trade off evaluation cost for accuracy by adjusting grid resolution.
>
>
> ---
> ## Framework Extendability
>
> Thank you for the suggestion. Our framework is naturally extendable to prune other structures beyond FFN neurons and attention heads. Since the genetic algorithm learns real-valued weights and is agnostic to the specific architectural units being pruned, it generalizes to any modular network component. This includes individual attention weights and even low-rank modules like LoRA.
>
>
> We conducted an experiment where we pruned individual attention weights rather than entire heads. For that, we use the same weighting from the genetic algorithm, but when creating subnetworks from the prunability score, we pruned individual attention weights rather than whole heads. However, while this is technically feasible, the practical speed-up is limited, as hardware accelerators cannot exploit sparsity at the fine-grained level of individual weights. In contrast, pruning entire heads allows structural simplifications that lead to realized speedups. Below, we report the linear classification accuracy for an AugReg ViT-B/16 backbone across different sparsity levels, comparing attention head pruning vs. individual attention weight pruning:
>
> | Sparsity | Ours (Attention Weights) | Ours (Attention Heads) |
> |----------|-----------------------|----------|
> | 0%       | 73.6          | 73.6        |
> | 10%      | 71.6        | 71.8  |
> | 20%      | 64.2          | 65.1  |
> | 30%      | 49.3            | 52.5 |
> | 40%      | 38.7        | 41.2   |
> | 50%      | 35.1     | 36.3   |
>
> We observe that pruning entire heads consistently outperforms finegrained pruning. This aligns with evidence that attention heads function as coherent computational units [4, 5], and pruning individual weights inside them can disrupt their effectiveness. Previous work that finds finegrained pruning to outperform structured pruning typically relies on finetuning to recover performance after pruning, which is not part of our method’s setup.
>
> [4] Michel, P., Levy, O., & Neubig, G. (2019). Are Sixteen Heads Really Better than One? NeurIPS 2019.
>
> [5] Voita, E., Talbot, D., Moiseev, F., Sennrich, R., & Titov, I. (2019). Analyzing Multi-Head Self-Attention: Specialized Heads Do the Heavy Lifting, the Rest Can Be Pruned. ACL 2019.
>
>
> ### PCA dimensionality.
> We ablated PCA dimensions to assess their effect on pruning quality. Increasing the PCA size from 192 to 384 led to a modest improvement in average linear classification accuracy (42.5% → 42.9%) at 50% sparsity on AugReg ViT-B/16, though with increased computational cost. Not using PCA at all resulted in a notable drop to 41.5%, indicating PCA helps both reduce computation and regularize the embedding space for better fitness evaluation. We chose 192 dimensions as a practical balance between performance and efficiency, making it the default setting for our experiments.
>
> ---
> ## Extreme sparsity levels:
> The trade-off between sparsity and performance is highly dependent on both model type and model size. We found that large-scale models—such as ViT-H DeiT-III with 630M parameters—can be pruned effectively with minimal performance loss at 40% sparsity (Table 9). Figure 10 further illustrates the relationship between performance degradation and GFLOPs for ViT-S, ViT-B, ViT-L, and ViT-H models from the DeiT-III family, pruned up to 60%. Notably, larger models tend to have more redundant parameters, making them more amenable to pruning.
>
> To quantify pruning efficiency, we introduce an efficiency score—defined as the percentage of GFLOPs that can be reduced while retaining at least 70% of the original performance. According to this metric, ViT-S allows a 40% GFLOPs reduction, while ViT-B and ViT-L achieve 56%, and ViT-H up to 60%. Additionally, as shown in Figure 4, different model architectures exhibit varying degrees of prunability. For instance, CLIP models demonstrate better tolerance to high sparsity levels compared to DINO models.
>
> Our primary focus is on pruning without any retraining or finetuning, driven by the increasing demand for efficient deployment of large foundation models across diverse downstream tasks. Under this constraint, we typically observe a significant performance drop beyond 50% sparsity. However, as demonstrated in our responses to Reviewers e4hx and ZHY4, this degradation can be nearly fully recovered through finetuning—bringing our method in line with state-of-the-art approaches in retraining settings.
>
> We will include a more detailed discussion of the trade-off between performance and compute reduction in the revised version of the paper. For additional insights, please refer to our response to Reviewer e4hx regarding performance prop point analysis.

---

> > ### Comment · Reviewer_69Bj · 2025-08-05
> >
> > Thank you for your detailed responses to my questions. After reviewing your clarifications, I have a better understanding of the method and its contributions. However, I still have some concerns regarding the time overhead associated with your approach.
> >
> > Scaling to larger models seems to introduce significant time costs, and this overhead appears not to be a one-time expense. Instead, it needs to be incurred for each model size, which could result in a linear increase in cost as more models of different sizes are required. This raises concerns about the scalability and efficiency of the approach in practical settings, particularly when dealing with a large range of model sizes.
> >
> > I will update my rating after further consideration, taking into account the feedback from other reviewers and the ongoing discussion.
> >
> > Thank you again for your thorough explanations.

---

> > > ### Author Response · Authors · 2025-08-05
> > >
> > > Thank you for taking the time to engage in the discussion!
> > >
> > > We’d like to clarify that our method is efficient in terms of overhead and, importantly, **does not grow linearly with the number of desired model variants**. Unlike methods such as SparseGPT and LLM Surgeon, which must be run separately for each target sparsity level, our **elastic pruning approach** produces all sparsity levels in a single run.
> > >
> > > This means that if one wished to obtain N pruned model variants from a single source model, our method would incur a **constant cost**, whereas other methods scale linearly with N. The apparent increase in runtime for larger models stems solely from the natural slowdown in forward passes for those models—not from algorithmic inefficiency.
> > >
> > > Below, we report the time to generate six sparsity levels (10%, 20%, 30%, 40%, 50%, 60%) across multiple model sizes. For our method each row requires **only a single pruning run**, whereas for other methods we sum the times for all six individual pruning runs for each sparsity.
> > >
> > > | Pruning time | Params | Blocks| Heads | Forward [img/s] |  Ours | SparseGPT | LLM Surgeon |
> > > |--|--|--|--|--|--|--|--|
> > > | **DeiT‑III ViT‑S/16** | 22 M | 12 | 6  |   1385     |2 min 35 s |          2 min 30 s  |    5 min 15s    |
> > > | **DeiT‑III ViT‑B/16** | 86 M | 12 | 12 |    434   |  2 min 55 s |        3 min 12 s     |   6 min 36 s    |
> > > | **DeiT‑III ViT‑L/16** | 304 M | 24 | 16 |   131    | 4 min 58 s |        8 min 6s   |    17min 5 s   |
> > > | **DeiT‑III ViT‑H/14** | 632 M | 32 | 16 |   48    | 11 min 04 s |        15 min 30s    |    32 min 37s   |
> > > | **EVA ViT-G/14** | 1B | 40 | 16  |    14    | 20 min 05s   |             37min 42 s  |    –   |
> > >
> > > Even when producing only six sparsity levels, our method is faster than SparseGPT and LLM Surgeon, and its scaling trend matches that of existing approaches. We will include these results in the final version. We hope this clarifies the remaining concern.

---

> > > > ### Author Response · Authors · 2025-08-08
> > > >
> > > > Thank you for engaging in the discussion. We addressed your concern regarding overhead by showing that our method scales with constant cost, as it performs single-shot elastic pruning across all sparsity levels. Does this resolve your concern, or is there anything else we can clarify? If not, we would be grateful if you could consider updating your score.

---

> > > > > ### Comment · Reviewer_69Bj · 2025-08-08
> > > > >
> > > > > Thank you for the detailed and thoughtful response. From the perspective of time overhead, the method proposed in this paper indeed exhibits lower overhead compared to SparseGPT and LLM Surgeon when generating six scale models. However, I do not observe evidence that SparseGPT and LLM Surgeon scale linearly with N. For the DeiT-III ViT-S/16 model, if generating six sparsity levels takes a total of 2 min 30 s, does this imply that generating a single model would only require 25 seconds, assuming the overhead grows linearly?
> > > > >
> > > > > Compared to the proposed method (2 min 35 s), I still have concerns regarding the additional overhead introduced by applying evolutionary algorithms to pruning, as this remains a limitation that cannot be overlooked, despite the authors' emphasis on its efficiency.

---

> > > > > > ### Author Response · Authors · 2025-08-09
> > > > > >
> > > > > > Thank you for engaging in the discussion. We are happy to clarify this concern:
> > > > > >
> > > > > > Yes, SparseGPT and LLM Surgeon prune each sparsity level independently, so total runtime grows approximately linearly with the number of targets (e.g., for ViT-Small: ~25 s per level, totaling ~2 min 30 s for six levels).
> > > > > >
> > > > > >
> > > > > > In contrast, our method optimizes for a **spectrum of sparsity levels in a single pruning run** (e.g., 10%, 30%, 50%, 60%), yielding a global importance ranking over prunable units. Any pruned model can then be produced instantly by thresholding this shared ranking—eliminating the need for repeated pruning runs.
> > > > > >
> > > > > >
> > > > > > Our method can also be run optimizing for a **single target sparsity**, resulting in significantly faster runtimes while maintaining performance. Below, we report **per-sparsity pruning times** across methods to ensure an apples-to-apples comparison:
> > > > > >
> > > > > > | **Pruning time per sparsity**                  | **Params** | **Blocks** | **Heads** | **Forward [img/s]** | **Ours** | **SparseGPT** | **LLM Surgeon** |
> > > > > > |---------------------------|------------|------------|-----------|---------------------|----------|----------------|------------------|
> > > > > > | **DeiT‑III ViT‑S/16**     | 22 M       | 12         | 6         | 1385                | **~24 s**    | ~25 s      | ~52 s            |
> > > > > > | **DeiT‑III ViT‑B/16**     | 86 M       | 12         | 12        | 434                 | **~26s**    | ~32 s      | ~66 s            |
> > > > > > | **DeiT‑III ViT‑L/16**     | 304 M      | 24         | 16        | 131                 | **~69 s**| ~81 s          | ~171 s           |
> > > > > > | **DeiT‑III ViT‑H/14**     | 632 M      | 32         | 16        | 48                  | **~135 s**| ~155 s        | ~326 s           |
> > > > > > | **EVA ViT-G/14**          | 1B         | 40         | 16        | 14                  | **~320s**   | ~377 s         | –                |
> > > > > >
> > > > > >
> > > > > > These results show that our method is **faster in runtime across all model sizes when pruning a single sparsity level**. After a one-time local Hessian computation, the evolutionary search uses **only forward passes**—no gradients or second-order derivatives—thanks to natural-gradient updates to evolve the search distribution efficiently. This design keeps our approach both effective and lightweight in practice.
> > > > > >
> > > > > >
> > > > > > But the key strength of our method is the ability to optimize for a sparsity spectrum and produce **any desired variant on-the-fly at deployment**, enabling both practical deployment and performance gains over prior work.

---

### Official Review · Reviewer_ZyhL · 2025-06-27

**Clarity:** 4
**Significance:** 3
**Originality:** 3
**Rating:** 5
**Confidence:** 4

**Summary:**

This paper introduces a structured pruning method for Vision Transformers (ViTs) that enables the creation of elastic inference models directly from pretrained models, eliminating the need for retraining. The proposed approach combines second-order sensitivity measures with evolutionary algorithm-based approximations. The authors demonstrate substantial improvements over several baseline methods across a range of image classification and segmentation tasks.

**Questions:**

- How sensitive is the performance of the evolutionary algorithm to the choice of initial parameters or random seeds?

- What impact would including first-order terms have on pruning performance, given that these terms are typically non-zero?

- How does the proposed pruning approach compare to random pruning when evaluated using KNN-based methods?

- The authors mention challenges in applying their pruning method to large models trained on large-scale datasets. What specifically were these challenges—reduced performance, computational inefficiency, or something else?

- Are there any limitations to applying this approach to Vision Transformers (ViTs), or is it equally effective for other model architectures?

**Ethical Concerns:**

["NO or VERY MINOR ethics concerns only"]

**Final Justification:**

This paper presents a novel structured pruning approach for ViTs that combines second-order sensitivity with evolutionary search, enabling elastic inference without retraining. The rebuttal addressed my key concerns by adding a complexity analysis, comparisons to finetuning-based baselines, and robustness studies, which significantly strengthen the contribution. Overall, I find the paper technically solid, well-motivated, and impactful for practical deployment, and I recommend acceptance.

**Limitations:**

yes

**Paper Formatting Concerns:**

No concerns

**Quality:**

3

**Strengths And Weaknesses:**

Strengths:

-To the best of my knowledge, the combination of gradient-based pruning with an evolutionary approximation of Hessian correlations is a novel and innovative approach.

-The method demonstrates strong performance gains across multiple benchmark datasets and various Vision Transformer architectures, including DINO, AugReg, and CLIP.

-It achieves effective pruning of Vision Transformers with minimal computational overhead, making it well-suited for practical deployment scenarios.

-The paper includes a thorough analysis of performance in relation to throughput and FLOPs, offering valuable insights into the trade-offs involved.

Weaknesses:

-The paper lacks an analysis of computational complexity, like done in SOSP (Second-order Structured Pruning). Additionally, SOSP and similar approaches are not mentioned in the related work section as viable alternatives for approximating the full Hessian in pruning (line of Approach using Gauss-Newton-Approximation also done in Eigendamage). Including this comparison (in terms of complexity) would help clarify the scalability of the proposed method.

-There is no comparison with supervised pruning methods that incorporate fine-tuning after pruning. Including such comparisons would strengthen the evaluation and address a current weakness of this work, which is that it only benchmarks against three other methods.

---

> ### Author Rebuttal · Authors · 2025-07-31
>
> Thank you for the valuable feedback and for recognizing the novelty of our approach, its strong performance across benchmarks, the effective pruning of Vision Transformers with minimal computational overhead, and our analysis of performance for throughput and FLOPs. Below, we address each of the raised questions:
>
> ---
> ## Analysis of computational complexity:
> We thank the reviewer for requesting a formal complexity analysis and for pointing us to SOSP and EigenDamage. Both papers will be added to the related‑work section as well as the analysis.
> The algorithm’s total cost is
> $$
> \mathcal{O}(N_D F)
> \+
> \mathcal{O}(T \lambda S N_S F')
> \+
> \mathcal{O}(T B^{2})
> \+
> \mathcal{O}(P \log P),
> $$
>
> where
>
> * **\\(\mathcal{O}(N_D F)\\)** computes the local diagonal Hessian via one forward–backward pass on \\(N_D\\) samples (\\(F\\) per pass).
> * **\\(\mathcal{O}(T \lambda S N_S F')\\)** in every generation of the xNES, we draw \\(\lambda\\) candidate masks, test each one at \\(S\\) sparsity targets on \\(N_S\\) images; \\(F'\\) is a forward‑only pass (feature extraction + PCA), so no back‑propagation is involved.
>
> * **\\(\mathcal{O}(T B^{2})\\)** updates the \\(B \\times B\\) covariance matrix \\( \Sigma \\) in xNES (FFN + head blocks), capturing global off‑diagonal structure.
> * **\\(\mathcal{O}(P \log P)\\)** sorts \\(P\\) prunability scores once to yield elastic subnetworks.
>
> The key speed‑up over SOSP stems from **zero backward curvature calls**: SOSP‑H requires one Hessian–vector product per structure and SOSP‑I builds a dense \\(S \times S\\) Gauss–Newton matrix, whereas we rely exclusively on forward inference and a lightweight covariance update \\(\mathcal{O}(T B^{2})\\) (contributing <0.01%).
>
> Runtime is therefore dominated by feature extraction.  With 50 xNES iterations and four sparsity targets, a single A100 prunes the entire DeiT‑III family in 2–11 min: ViT‑S(2 : 35), ViT‑B(2 : 55), ViT‑L(4 : 58), and the 32‑block ViT‑H(11:04).  The quadratic covariance term remains negligible throughout, demonstrating efficient scalability even for the largest ViTs.
>
> ---
> ## Finetuning comparison:
> Thank you for highlighting this. Our work intentionally targets the supervision-free, no-finetuning pruning setting, motivated by real-world deployment constraints (e.g., multi-task inference, edge applications, or frequent model updates), where finetuning is often infeasible or expensive. In such cases, retraining must be repeated for each sparsity level, dataset, and task. For instance, finetuning a ViT-B on ImageNet takes ~3 days on a single A100 GPU, whereas finetuning a ViT-H requires over 12 days, making scaling challenging.
>
>
> That said, we agree that comparing with supervised finetuning-based methods is important to analyze the effectiveness of our approach. We have now added comparisons to several state-of-the-art pruning methods that assume full supervision and post-pruning finetuning, including SN-Net [4], NViT [1], LPViT [2], and SAViT [3]. As shown below, our method is highly competitive even in the finetuning setting: at 50% sparsity, we achieve 81.6% ImageNet top-1 accuracy after finetuning, nearly matching the unpruned baseline (81.8%) while offering a **76% inference speedup**.
>
> | Method     | ImageNet Top-1 | Avg. k-NN | Avg. Linear | Finetuning Epochs | Speedup     |
> |------|---|-|--|--|-------------|
> | Unpruned   | 81.8   | 75.8      | 78.5        | –      | 0%          |
> | Ours       | 81.6   | 75.0      | 75.1   | 100    | 76% (A100)  |
> | NViT [1]    | 83.3 | 73.7      | 72.0     | 300         | 75% (A100)  |
> | LPViT [2]   | 80.6     | –         | –     | 300        | 79% (V100)  |
> | SAViT [3]   | 82.6  | –         | –      | 300   | 55% (V100)  |
> | SN-Net [4]  | 80.0   | 70.2  | 71.4    | 100   | 83% (3090)  |
>
> We will include the full finetuning results for all sparsities in the updated manuscript. While our primary contribution is enabling **fast, label-free pruning without retraining**, we show that our method also remains **finetuning-compatible and competitive**.
>
> [1] Yang et al., *Global Vision Transformer Pruning with Hessian-Aware Saliency*, CVPR 2023
>
> [2] Xu et al., *LPViT: Low-Power Semi-structured Pruning for Vision Transformers*, ECCV 2024
>
> [3] Zheng et al., *SAViT: Structure-Aware Vision Transformer Pruning*, NeurIPS 2022
>
> [4] Pan et al., *Stitchable Neural Networks*, CVPR 2023
>
> ---
> ## Method robustness to initialization and random seeds
>
> We investigate the sensitivity of our pruning approach to variations in random seeds and the initialization parameters of the evolutionary algorithm.
>
> #### **Random Seeds**
> Table 7 in the Appendix reports k-NN and linear classification accuracy (mean ± standard deviation) across three random seeds for an AugReg ViT-B/16 backbone pruned to 10%, 30%, and 50% sparsity. We observe that performance variance is minimal at low sparsity levels (e.g., an average of 0.2% at 10%) and increases modestly at higher sparsity levels (up to 1-2%). Some datasets, such as EuroSAT, remain stable across seeds, while others, like Oxford-IIT Pets, are more sensitive, particularly in the k-NN setting. This suggests that our method is generally stable, with expected variability under more aggressive pruning.
>
> #### **xNES Iterations (\\( T \\))**
> As shown in Table 3, increasing the number of xNES iterations from 50 to 250 and then to 500 results in an approximate 2% performance improvement, indicating that the method converges effectively even with relatively few update steps.
>
> #### **Initialization of Covariance Matrix (\\( \Sigma \\))**
> We found that initializing the xNES covariance matrix \\( \Sigma \\) using CKA similarity scores between architectural components (e.g., attention heads and MLP blocks) leads to faster convergence and significantly improved pruning outcomes. As shown in Table 8, appendix, this initialization improves both k-NN and linear accuracy across all datasets, for instance, k-NN accuracy on ImageNet-1k increases from 19.5% to 40.4%, and on Oxford-IIT Pets from 27.6% to 58.0%, highlighting the effectiveness of structure-aware initialization.
>
> Our pruning method thus shows strong robustness across random seeds, xNES iteration counts, and initialization of \\( \Sigma \\).
>
> ---
> ## Impact of including First-Order term
> Thanks for pointing this out. We explored adding explicit first-order information to the pruning score in addition to the local Hessian (the diagonal approximation of the Hessian). To ensure compatibility with our genetic algorithm, which constrains weights to be non-negative , we use the absolute values of the gradient terms.
>
> Below, we report k-NN and linear classification accuracy (%) on an AugReg ViT-B/16 model across sparsity levels:
>
> | Sparsity | k-NN (Local Hessian) | k-NN (Local Hessian + Gradient) | Linear (Local Hessian) | Linear (Local Hessian + Gradient) |
> |---|-----|-------|---|-----|
> | 0%       | 73.6                 | 73.6        | 73.6    | 73.6     |
> | 10%      | 72.8                 | 72.7     | 78.1     | 78.6       |
> | 20%      | 68.2                 | 65.7      | 74.5     | 71.8   |
> | 30%      | 60.7                 | 57.4             | 67.3      | 64.8        |
> | 40%      | 47.3                 | 42.6           | 53.2                   | 46.9      |
> | 50%      | 39.3                 | 38.3         | 42.5       | 40.4           |
>
> The gradient term introduces noise from first-order optimization dynamics, which appears to distort the fitness signal used during pruning. In contrast, the local Hessian alone provides a more stable and reliable criterion, which is why we retain it as our default.
>
> ---
> ## Comparison to Random Pruning
> The table below reports average k-NN accuracy (%) across seven benchmark datasets for an AugReg ViT-B/16 model pruned using our method versus random pruning. Both methods start from the same unpruned baseline.
>
> | Sparsity | 0%   | 10%  | 20%  | 30%  | 40%  | 50%  | 60%  |
> |---|-------|-------|-------|-------|-------|-------|-------|
> | Ours     | 73.6  | 72.8  | 68.2  | 60.7  | 47.3  | 39.3  | 35.9  |
> | Random   | 73.6  | 66.9  | 47.7  | 37.6  | 33.3  | 30.2  | 26.9  |
>
> Our method consistently outperforms random pruning at all sparsity levels, with particularly large gains at moderate-to-high sparsity (e.g., +21% at 30% sparsity). These results demonstrate the value of structure-aware pruning: unlike random pruning, our approach preserves semantically meaningful representations that remain effective for downstream tasks without requiring finetuning.
>
> ---
> ## Challenges with large models:
>
> Large Vision Transformers like ViT-H are pretrained on hundred-million–scale datasets, which leads to a much flatter loss landscape. As a result, the xNES optimizer requires roughly 75–100 warm-up iterations just to explore the landscape before the loss begins to decrease. To obtain a pruning mask of comparable quality, we therefore need to **extend the search horizon \\( T \\)**.
>
> While this increases the number of optimization steps, we find that larger models are better to prune, as they contain more redundant parameters. For example, ViT-H can be pruned to reduce **40% of GFLOPs with virtually no performance loss** (see Table 9 in the appendix).
>
> ---
> ## Extendability of our approach
> While our paper focuses on ViTs, the method is not architecture-specific. It requires only (1) a gradient-based loss, (2) structural units to prune (e.g., layers, blocks, filters), and (3) a fitness function for the genetic algorithm. This general framework makes it applicable to other architectures like CNNs, and domains such as language modeling or audio. For instance, in language models, one could prune attention heads and MLP blocks using a next-token prediction loss.

---

> > ### Comment · Reviewer_ZyhL · 2025-08-07
> >
> > Thank you for the detailed and thoughtful rebuttal. I’m satisfied with the authors’ clarifications and additions—especially the formal complexity analysis, new comparisons to finetuning-based baselines, and robustness investigations. The responses address the key concerns well and further strengthen the paper’s contribution.

---

### Official Review · Reviewer_e4hx · 2025-06-28

**Clarity:** 3
**Significance:** 2
**Originality:** 2
**Rating:** 4
**Confidence:** 3

**Summary:**

Vision foundation models are limited by fixed architectures that force suboptimal deployment decisions under real-world computational constraints. The paper presents a post-pretraining structured pruning method that enables elastic inference across different compute budgets by combining gradient information with cross-network structure correlations. These are approximated through evolutionary algorithms, requiring no labeled data or retraining. The approach significantly outperforms existing methods on DINO and AugReg models across various sparsity levels. It generates flexible models in under five minutes on an A100 GPU that can be dynamically adjusted to any computational budget.

**Questions:**

* It is clear that at 50% pruning you nicely improve over SOTA. But why does 50% matter? In my understanding, a key goal of novel pruning methods is to move the drop point, i.e., the point at which the model drastically loses in terms of performance. In this critical region, the paper on shows very minimal or even no improvements over the state of the art for some benchmark setting (Figure 2 a and b). I lack a discussion of this.

* For others (e.g. Figure 3), they show nice improvements over but only over simple baseline (activations, LAMP) and do not compare against SOTA baselines. This makes it very hard for me to judge the contributions. Please close these gaps.

* I suggest showing both the baselines and the authors' approach with and without post-pruning fine-tuning. Fine-tuning is affordable in many use-cases, and I think it is important to evaluate how "fine tunable" an approach is. Especially, as most baselines are designed for fine-tuning.

* In some settings the authors report throughput, in others not. Why?

* "Our method improves over baselines, while not using labels:" In my understanding, the baseline also do not use labels

* "Our method retains performance for downstream tasks:" This statement feels a bit too strong, at least for the high pruning rates.

This paper is very interesting, but I lack:
* motivation on the need for approaches without any fine-tuning
* a focus on the critical regions of pruning
* fully clean evaluation

**Ethical Concerns:**

["NO or VERY MINOR ethics concerns only"]

**Final Justification:**

I thank the authors for their quick response and the new insights. After also rechecking the other reviews, I increase my score to reflect the quality of their new results.

**Limitations:**

yes

**Quality:**

3

**Strengths And Weaknesses:**

Strength:
* well-written paper
* Important field: elastic models
* nice: elastic models without the need for fine/tuning / retraining.
* detailed supplementary material / appendix.

Weaknesses:
* limited improvements or even none in the critical accuracy regions in some settings
* in other settings: lacks of SOTA baselines
* strong focus on pruning without any finetuning, while many SOTA works include fine-tuning.

---

> ### Author Rebuttal · Authors · 2025-07-31
>
> Thank you for the insightful feedback and for recognizing the importance of elastic models, specifically in their ability to operate without the need for finetuning, and for the well-presented paper and appendix. Below, we address each of the raised questions in detail:
>
> ---
> ## Motivation for pruning without the need for finetuning:
> Our work focuses on post-training structured pruning without finetuning to address increasingly common deployment and research needs:
> - Generalist, multi-task use cases: Many models today (e.g., CLIP, DINO, or AugReg) serve broad roles across diverse tasks, as we also show in our evaluation across seven bechmarks. However, finetuning for each downstream target might be infeasible due to proprietary constraints, latency budgets, or lack of labeled data. Robust, general-purpose pruning is thus essential.
> - Preserving robustness and generalization: We observe that pruning+finetuning on a narrow dataset (e.g., CIFAR-10) can reduce generalization (see e.g., accuracy drop of 3.1% from 64.2 to 61.1% via k-NN). This highlights that pruning **without** retraining can better preserve the inherent robustness encoded in foundation models.
> - Fast iteration & model churn: In rapidly evolving domains, frequent model updates make repeated fine-tuning costly. Our one-shot, retraining-free method offers a scalable alternative with strong performance across benchmarks.
> - Scientific insight: By avoiding finetuning, we can more directly analyze how pruning alters the learned representations. This links to ongoing foundational work on interpretability and the role of structural representations.
>
> We will clarify these points more explicitly in the future version to help readers better understand the impact and scope of our work.
>
> ---
> ## Drop point analysis:
> We apologize if the plots (e.g., Figure 2) were not sufficiently clear in illustrating our improvements near the drop point. To clarify, we provide a detailed table below that demonstrates consistent gains over FPTP and LLM surgeon in the 30, 40, and 50% sparsity range, on average across the seven benchmark datasets, while also using label supervision for pruning an AugReg ViT-B/16 backbone:
>
> | Sparsity | Method       | k-NN Avg. | Linear Avg. |
> |----------|--------------|-----------|-------------|
> | **30%**  | **Ours**         | **63.7**      | **70.4**        |
> |          | FPTP         | 60.7      | 68.3        |
> |          | LLM Surgeon  | 54.6      | 59.8        |
> |----------|--------------|-----------|-------------|
> | **40%**  | **Ours**         | **51.2**      | **57.7**        |
> |          | FPTP         | 44.7      | 51.5        |
> |          | LLM Surgeon  | 34.0      | 36.3        |
> |----------|--------------|-----------|-------------|
> | **50%**  | **Ours**         | **41.0**      | **43.6**        |
> |          | FPTP         | 34.5      | 36.3        |
> |          | LLM Surgeon  | 32.5      | 28.5        |
>
> These results show that our method significantly outperforms FPTP and LLM surgeon, already before the 50% mark, thus delaying the drop point. In the response below, we show that we can recover from such drops via finetuning. We will better visualize and provide these results in the updated version.
>
> ---
> ## Baselines & full finetuning results:
> Thank you for the suggestion. Our work focuses on pruning **without any retraining or finetuning**, motivated by the growing need to deploy large foundation models efficiently across diverse downstream tasks **without the computational cost or complexity of finetuning** (see earlier response).
>
> While finetuning can help recover accuracy after pruning, [1, 3] have shown that full retraining often not only recovers the original performance but even improves it. However, finetuning must be repeated for every sparsity level and each downstream task, which can be computationally expensive. For example, finetuning ViT-B for 100 epochs on ImageNet takes roughly 3 days on a single A100 GPU, and ViT-H requires more than 12 days. This poses a significant bottleneck in multi-task or dynamic deployment scenarios, or when models are frequently updated. Our work targets this more challenging but practically relevant scenario, requiring only about 5 minutes for pruning on a single A100 and no label supervision. Because of this, fewer directly comparable baselines exist, as most prior work assumes finetuning.
>
> That said, we also demonstrate our method’s **finetunability**: pruning 50% of DeiT ViT-B and finetuning on ImageNet-1k achieves 81.5% top-1 accuracy, close to the unpruned baseline of 81.8% while delivering a 76% inference speedup. Our results are competitive with state-of-the-art pruning methods. We will include a comprehensive table with all sparsity levels in the updated manuscript.
>
> | Method  | ImageNet Top-1 | Average k-NN | Average Linear | Finetuning Epochs | Speedup  |
> |---------|----------------|--------------|----------------|--------|------------------|
> | Unpruned |  81.8   |    75.8         | 78.5     | –  | 0%     |
> | Ours    | 81.6          | 75.0         | 75.1          | 100     | 76% (A100)     |
> | NViT [1]   | 83.3           | 73.7         | 72.0           | 300    | 75% (A100)     |
> | LPViT [2]  | 80.6          | –         | –            | 300    | 79% (V100)     |
> | SAViT [3]  | 82.6          | –          | –            | 300    | 55% (V100)     |
> | SN-Net [4] | 80.0           | 70.2        | 71.4          | 100    | 83% (RTX 3090) |
>
> Thank you for encouraging this experiment which demonstrates strong performance gains when finetuning is employed. We will add this experiment for all sparsities to our updated version.
>
> [1] Huanrui Yang et al. Global Vision Transformer Pruning with Hessian-Aware Saliency. CVPR 2023.
>
> [2] Kaixin Xu et al. LPViT: Low-Power Semi-structured Pruning for Vision Transformers. ECCV 2024
>
> [3] Chuanyang Zheng et al. SAViT: Structure-Aware Vision Transformer Pruning via Collaborative Optimization. NeurIPS 2022
>
> [4] Zizheng Pan, Jianfei Cai, Bohan Zhuang: Stitchable Neural Networks. CVPR 2023
>
> ---
> ## Throughput vs GFLOPS:
> In Figure 2 (a, b), we report GFLOPs as floating-point operations per inference pass, as the LLM Surgeon and FPTP mask out pruned structures rather than effectively removing them, making it non-trivial to realize the theoretical speedups in practice. Therefore, we do not report throughput for these baselines and compare them to our method only in terms of GFLOPs. In contrast, our method and NViT produce pruning patterns that are hardware-friendly and straightforward to realize, so we report throughput for these methods as it better reflects actual runtime performance.
>
> ---
> ## Labels for pruning:
>
> We appreciate the reviewer’s comment and would like to clarify. While we do offer a cross-entropy (CE) variant of our method for comparison, our primary approach does not require any labels, instead relying on a self-supervised loss (Eq. 5) during pruning.
> In contrast, most baselines, including FPTP, LLM surgeon, and NViT, rely on supervised cross-entropy losses and assume access to a linear classification head during pruning. This limits both their flexibility and generalization, especially for pretrained models where such heads are unavailable; thus, we cannot report a comparison with those methods for DINOv1, DINOv2, and CLIP.
> We outperform these baselines even without using labels, showing that meaningful structure-aware pruning is possible in a fully unsupervised setting (see Table 1).
>
> ---
> ## Retaining performance claim:
> Thank you for pointing this out. We agree that the phrasing “retains performance for downstream tasks” may be too strong, especially at very high sparsity levels. While we do observe strong retention even at high sparsities for big models (e.g., Table 9 in the appendix; DeIT-III ViT-H/14), the drop in performance at extreme pruning rates is indeed dataset- and model-dependent.
> We will revise the wording in the future version to more accurately reflect this nuance, emphasizing that our method preserves downstream performance well in the practical sparsity range (e.g., 30-40%), which is also where it most consistently outperforms baselines.

---

> ### Comment · Reviewer_e4hx · 2025-08-02
>
> I thank the authors for the time and effort to answer my (and the other reviewers) questions. After reading the other reviews and the comments to all reviews, this answers most of my questions. Moreover, it is very interesting to see that the proposed method also works well with fine-tuning (although it was not explicitly designed for this). I would like to note that these are baseline with "normal" fine-tuning and not ones that focus on minimal fine-tuning, i.e., one shot-pruning.
>
> Some concerns remain. The strongest one is still the lack of comparison to SOTA approaches and baselines, both in the discussion and in the experimental evaluation. For example, in the field of single-shot / one-shot pruning, we have (among others):
> * H. Kohama, H. Minoura, T. Hirakawa, T. Yamashita and H. Fujiyoshi, "Single-Shot Pruning for Pre-trained Models: Rethinking the Importance of Magnitude Pruning," 2023 IEEE/CVF International Conference on Computer Vision Workshops (ICCVW)
> * Zhengyan Zhang, Fanchao Qi, Zhiyuan Liu, Qun Liu, Maosong Sun, Know what you don't need: Single-Shot Meta-Pruning for attention heads, AI Open, Volume 2, 2021, Pages 36-42, ISSN 2666-6510, https://doi.org/10.1016/j.aiopen.2021.05.003.
> * Elias Frantar and Dan Alistarh. 2023. SparseGPT: massive language models can be accurately pruned in one-shot. In Proceedings of the 40th International Conference on Machine Learning (ICML'23), Vol. 202. JMLR.org, Article 414, 10323–10337.
> * Preserving Deep Representations In One-Shot Pruning: A Hessian-Free Second-Order Optimization Framework
> https://arxiv.org/abs/2411.18376
>
> Plus numerous works also from the pre-transformer time.
>
> I note that many of these works focus on minimal fine-tuning (instead of no fine-tuning as the authors do). However, their focus is too similar to ignore them. Especially, as for some the fine-tuning effort is very minimal, and they achieve very competitive results. Here, a direct comparison of performance, e.g., accuracy, parameters, and throughput, versus the cost of fine-tuning is important.

---

> > ### Author Response · Authors · 2025-08-04
> >
> > We thank the reviewer for their feedback and for recognizing the effort we put into the responses. We also appreciate the recognition that our method performs well even under standard finetuning, despite not being designed for it. We agree that the field of one-shot pruning includes several relevant works, and we address each of the suggested papers in detail below. We will incorporate this discussion and results into the updated paper version to ensure a more complete comparison with state-of-the-art approaches.
> >
> > | Method | 0%   | 10%  | 20%  | 30%  | 40%  | 50%  | 60%  |
> > |-|-|--|-|-|-|--|-|
> > | Kohama et al. [3] | 73.6 | 72.2 | 66.7 | 54.8 | 40.6 | 36.1 | 32.1 |
> > | SparseGPT [5] | 73.6 | 65.9 | 61.1 | 54.7 | 45.8 | 38.6 | 34.1 |
> > | Ours   | 73.6 | **72.6** | **70.6** | **63.7** | **51.2** | **41.0** | **35.4** |
> >
> > We compare with previous works  [3, 5]  which allow a direct comparison for structured pruning without retraining for any of the methods for AugReg ViT-Base using linear evaluation. We report the average across all six datasets, following the main paper. These results show that we **outperform state-of-the-art works such as SNIP [1] and SparseGPT [5]** for pruning on ViT-B using linear evaluation. Note that our method does the pruning in a  **single shot for all sparsities levels** (elastic pruning) while SparseGPT and SNIP prune for each specific sparsity ratio, thus our method is more efficient. We also discuss our approach with mentioned works:
> >
> > - Kohama et al. [3] (ICCVW 2023) builds on SNIP [1] by extending the objective with a squared magnitude pruning term (Eq. 4). We compare their method with ours for structured pruning without finetuning using  \\( \alpha  = 0.01 \\) which yielded the best performance. Our results show that our method significantly improves across all sparsity levels.
> >
> > - Zhang et al. [4] (AI Open 2021) prune attention heads in LLMs using a trained meta-pruner followed by finetuning. While this supports our observation that whole-head pruning is effective, their method is architecture-specific. In contrast, ours is **architecture-agnostic**, pruning arbitrary network structures to much higher sparsity levels without the need to train a module. Thus, we do not directly compare due to differing scopes and targets.
> >
> > - SparseGPT [5] (ICML 2023) is the closest baseline in our setting. It is a strong one-shot method designed for unstructured or semi-structured pruning in LLMs without finetuning. We compare against them using the official code implementation for structured pruning in ViTs, as acceleration gains cannot be effectively realized in unstructured pruning.  Our results show consistently improvements especially for low sparsity levels.
> >
> > - SNOWS [6] (ICLR 2025) proposes a Hessian-free second-order optimization technique that computes approximate Newton steps without explicitly forming the Hessian matrix. The method operates on top of existing pruning masks, refining them to improve performance. While it has demonstrated gains when combined with magnitude-based pruning or SparseGPT, it is not a standalone pruning method, as it does not generate masks independently. Given its role as an enhancement rather than a pruning strategy, we do not include it as a direct baseline. Nonetheless, we anticipate that integrating SNOWS into our framework will yield similar benefits.
> >
> > - SNIP [1] & GraSP [2] are foundational works that inform [3], which we already compare against. Nonetheless, we will ensure that [1] and [2] are cited appropriately to reflect the historical context.
> >
> >  For context, we also include the GFLOPs for each sparsity.
> >
> > | GFLOPs   | 0%   | 10%  | 20%  | 30%  | 40%  | 50%  | 60%  |
> > |-|--|--|--|--|-|-|-|
> > | Kohama et al. [3] | 35.7 | 31.8 | 28.3 | 24.9 | 21.3 | 17.9 | 14.3 |
> > | SparseGPT [5] | 35.7 | 31.7 | 28.4 | 25.0 | 21.6 | 18.2 | 15.0 |
> > | Ours| 35.7 | 31.8 | 28.3 | 24.9 | 21.3 | 17.9 | 14.3 |
> >
> > We will add these results and expand experiments for the final version. We hope we have clarified this remaining concern; if there are no remaining questions, we would kindly ask the reviewer to update the score.
> >
> > [1] Namhoon Lee, Thalaiyasingam Ajanthan, Philip H. S. Torr: SNIP: single-Shot Network Pruning based on Connection sensitivity. ICLR 2019.
> >
> > [2] Chaoqi Wang, Guodong Zhang, and Roger Grosse. Picking winning tickets before training by preserving gradient flow. ICLR 2020.
> >
> > [3] H. Kohama, H. Minoura, T. Hirakawa, T. Yamashita and H. Fujiyoshi, "Single-Shot Pruning for Pre-trained Models: Rethinking the Importance of Magnitude Pruning," 2023 ICCVW
> >
> > [4] Zhengyan Zhang, Fanchao Qi, Zhiyuan Liu, Qun Liu, Maosong Sun, Know what you don't need: Single-Shot Meta-Pruning for attention heads, AI Open, Volume 2, 2021, Pages 36-42.
> >
> > [5] Elias Frantar and Dan Alistarh. 2023. SparseGPT: massive language models can be accurately pruned in one-shot. ICML 2023
> >
> > [6] Preserving Deep Representations In One-Shot Pruning: A Hessian-Free Second-Order Optimization Framework. ICLR 2025.

---

> > > ### Comment · Reviewer_e4hx · 2025-08-05
> > >
> > > I thank the authors for their quick response and the new insights. After rechecking the other reviews, I will increase my score to reflect the quality of their new results.

---

> > > > ### Author Response · Authors · 2025-08-06
> > > >
> > > > We are thankful to the reviewer for acknowledging our new insights that made the paper stronger and updating the score.

---

### Note · Authors · 2025-08-14

We thank the reviewers for a constructive discussion that strengthened the paper and resolved core concerns.
## Key Updates
- **Expanded baselines**: Added one-shot pruning (Kohama et al., SparseGPT) and finetuning-based SOTA (NViT, LPViT, SAViT, SN-Net). Our method outperforms one-shot approaches and—when optionally finetuned—matches SOTA finetuning results. Crucially, a single run produces all sparsity levels, avoiding repeated pruning runs.
- **Scalability:** Complexity analysis shows the search is strictly forward-only, with xNES covariance updates adding <0.01% overhead and runtime growing proportionally with model size. A single pruning run generates all sparsity levels faster than SparseGPT and LLM Surgeon, and even for a single target sparsity, our method is consistently faster across ViT-Small to ViT-Giant.
- **Robustness**: Ablations over seeds, PCA dimensionality, and xNES iterations confirm stability and predictable trade-offs.
- **Hessian–xNES connection**: Added theoretical and intuitive justification for using inverse xNES covariance as a global Hessian proxy, with literature support.
- **Drop-point performance**: In the critical 30–50% sparsity range, we outperform FPTP and LLM Surgeon, delaying the accuracy collapse point.
## Reviewer Outcomes
- **e4hx** — Concerns resolved; score increased after expanded baselines, drop-point analysis, and finetuning results.
- **ZyhL** — Maintained an accept rating and expressed satisfaction after adding complexity analysis, finetuning comparisons, and robustness ablations.
- **69Bj** — Overhead and scalability concerns resolved with per-sparsity runtime data showing faster performance than SparseGPT/LLM Surgeon; score updated, no further objections.
- **ZHY4** — Clarity on Hessian–xNES integration, SN-Net comparison, and GFLOPs definition; finetuning and LLM applicability clarified; no further objections.
## Bottom Line
The paper introduces a **single-shot, label-free, retraining-free** pruning method that generates an elastic family of ViT subnetworks in minutes, competitive with or exceeding SOTA. It avoids repeated runs per sparsity, scales to billion-parameter models, and demonstrates robustness. All substantive concerns were resolved, with two reviewers updating/increasing scores and the others confirming satisfaction after clarifications.

---

### Decision · Program_Chairs · 2025-09-17

**Decision:**

Accept (poster)

**Comment:**

The authors propose a structured pruning method to enable elastic inference across a range off pre-defined sparsity levels, the primary novelty of the method is to use an approximate decomposition of the Hessian, both intra-block and inter-block, to inform the saliency criteria for the pruning method. The reviewers all agreed the proposed methodology was novel and well motivated. Most of the initial concerns of the reviewers were resolved in a robust discussion between the reviewers and authors. The main points of concern for the reviewers that appear to remain post-rebuttal are the scalability/computational efficiency of the method (and specifically if it is a constant cost), and the applicability of the proposed method to LLMs.

Having read through the discussion I believe the authors did address both of these concerns adequately, and I do not see these concerns as being sufficient to reject what appears to otherwise be a robust method, evaluation and writeup.